# Dependency of the impacts of geoengineering on the stratospheric sulfur injection strategy – Part 2: How changes in the hydrological cycle depend on injection rates and model?

Anton Laakso[1], Daniele Visioni[2], Ulrike Niemeier[3], Simone Tilmes[4], and Harri Kokkola[1]

[1]Finnish Meteorological Institute, Atmospheric Research Centre of Eastern Finland, Kuopio, 70200, Finland
[2]Department of Earth and Atmospheric Sciences, Cornell University, Ithaca, NY 14850, USA
[3]Max Planck Institute for Meteorology, Bundesstr. 53, 20146 Hamburg, Germany
[4]National Center for Atmospheric Research, Boulder, CO 80307, USA

**Correspondence:** Anton Laakso (anton.laakso@fmi.fi)

**Abstract.** This is the second of two papers where we study the dependency of the impacts of stratospheric sulfur injections on the used model and injection strategy. Here, aerosol optical properties from simulated stratospheric aerosol injections using two aerosol models (modal scheme M7 and sectional scheme SALSA), as described in Part 1, are implemented consistently into EC-Earth, MPI-ESM and CESM Earth System Models to simulate the climate impacts of different injection rates ranging from 2 to 100 Tg(S)yr$^{-1}$. Two sets of simulations were simulated with the three ESMs: 1) Regression simulations, where abrupt change in $CO_2$ concentration or stratospheric aerosols over preindustrial conditions were applied to quantify global mean fast temperature independent climate responses and quasi-linear dependence on temperature and 2) equilibrium simulations, where radiative forcing of aerosol injections with various magnitudes compensate the corresponding radiative forcing of $CO_2$ enhancement to study the dependence of precipitation on the injection magnitude; the latter also allow to explore the regional climatic responses. Large differences in SALSA and M7 simulated radiative forcings in Part 1 translated into large differences in the estimated surface temperature and precipitation changes in ESM simulations: e.g. an injection rate of 20 Tg(S)yr$^{-1}$ in CESM using M7 simulated aerosols led to only 2.2 K global mean cooling while EC-Earth - SALSA combination produced 5.2 K change. In equilibrium simulations, where aerosol injections were utilized to offset the radiative forcing caused by an atmospheric $CO_2$ concentration of 500 ppm, the decrease in global mean precipitation varied among models, ranging from -0.7% to -2.4% compared to the preindustrial climate. These precipitation changes can be explained by the fast precipitation response due to radiation changes caused by the stratospheric aerosols and $CO_2$ because global mean fast precipitation response is shown to be negatively correlated with global mean atmospheric absorption. Our study shows that estimating the impact of stratospheric aerosol injection on climate is not straightforward. This is because the simulated capability of the sulfate layer to reflect solar radiation and absorb LW radiation is sensitive to the injection rate as well as the aerosol model used to simulate the aerosol field. These findings emphasize the necessity for precise simulation of aerosol microphysics to accurately estimate the climate impacts of stratospheric sulfur intervention. This study also reveals gaps in our understanding and uncertainties that still exist related to these controversial techniques.

# 1 Introduction

One of the most studied Solar Radiation Modification (SRM) techniques is Stratospheric Aerosol Intervention (SAI), which has the intent of producing a layer of aerosols that reflects solar radiation back to space. Such techniques could artificially decrease the radiative imbalance caused by increased greenhouse gas (GHG) emissions and in theory, maintain radiative balance. However, in this theoretical case, all impacts would not be compensated. As GHGs suppress the outgoing long-wave (LW) radiation, SAI compensates for GHG induced radiative imbalance by altering mostly solar shortwave (SW) radiation. The magnitude of SRM could be adjusted to compensate GHG induced radiative flux change at top of the atmosphere, but not without changes in atmospheric energy budget as spatial and temporal structure of SW radiative fluxes differs from LW fluxes in the atmosphere. Thus, this would lead to several consequences such as a decrease in global mean precipitation and unevenly distributed temperature chance compared to climate without increase in $CO_2$ and SAI(e.g., Visioni et al. (2021); Laakso et al. (2020); Tilmes et al. (2013)). The extent of these impacts is influenced not only by the level of GHG increase in the atmosphere but also by the interaction of aerosol fields with SW and LW radiation. This interaction is further dependent on the aerosols' optical properties, which, as demonstrated in Laakso et al. (2022) study, are closely associated with the modeling of aerosol microphysics in climate models.

Most studies simulate injections of $SO_2$ for SAI. In this imitation of large volcanic eruptions, injected $SO_2$ oxidises to sulfuric acid and then either forms new particles or condenses on existing ones. Radiative properties of sulfate aerosols depend strongly on the size of these aerosols and thus are sensitive to ambient conditions during injections (background conditions and injection strategy). The sensitivity to microphysical processes (nucleation, coagulation, condensation) in climate models dependes very much on how such processes are modelled. Several studies on SAI using $SO_2$ injections show that for a fixed injection area, radiative forcing efficiency (i.e. radiative forcing/injection rate) decreases with a larger magnitude of injections (Heckendorn et al., 2009; Pierce et al., 2010; Niemeier et al., 2011). However, the magnitude of this reduction in the forcing efficiency and predicted radiative forcing are considerably different between studies and models. In Laakso et al. (2022) (from now on referred as Part 1), we simulated different injection rates using both a sectional aerosol model SALSA and a modal model M7 within the same climate model, showing that there are indeed significant differences in radiative forcings of SAI depending on how aerosol microphysics were simulated. In a case of continuous $SO_2$ injections to the Equator with injection rates of 1-100 Tg(S)yr$^{-1}$ produced 88 %–154 % higher global mean all-sky net radiative forcing when simulated with SALSA compared to M7.

In the case of SAI with sulfate, injected aerosols would not only scatter solar radiation, but also absorb SW and LW radiation. In Part 1 we showed that while SW radiative forcing (negative, i.e cooling impact) is saturating considerably with the injection rate, the relation of LW radiative forcing (positive, i.e warming impact) and injection rate is more linear (Niemeier and Timmreck, 2015). This means that in larger injection rates the contribution of LW radiation to total radiative forcing becomes larger: In SALSA simulations LW radiation forcing compensated for between 10% to 28% of the SW radiative forcing with injection rate of 1-100 Tg(S)yr$^{-1}$ while M7 simulation the range was 24-57%. This also has implications on how these radiative forcings translate to climate impacts since, as a side effect, aerosols are absorbing radiation and are warming the

atmosphere. The impact of LW absorption becomes stronger if larger injection rates are applied. This is also linked to how aerosols are modelled: in Part 1 SW radiative forcing was 45%–85% higher and LW radiative forcing was 24%-40% lower in simulations with SALSA than in M7. This indicates that there would be significant differences in the simulated climate responses depending on how the aerosols are simulated. The situation is further complicated by the lack of clear criteria for selecting the appropriate aerosol model. Observations following the 1991 Pinatubo eruption have frequently been utilized as a benchmark for evaluating models' ability to simulate stratospheric aerosols. However, a single sulfur injection, as in the case of Pinatubo, differs significantly from continuous injections in case of SAI. Notably, there is a minimal difference between the M7 and SALSA model results in the simulations of the Pinatubo eruption, as detailed in (Kokkola et al., 2018). Simulations using the M7 model were 60% faster than those with SALSA, but there were some numerical limitations associated with the modes in M7, which restricted the aerosols from achieving an optimal size range for effectively scattering radiation, as noted in Laakso et al. (2022). However, the performance of the M7 results is also sensitive to the configuration of the modes, making it difficult to predict which setup will perform well, as the performance depends on the simulated case (i.e volcanic eruption vs. SAI, different injection strategies for SAI).

Changes in atmospheric radiation have a direct impact also on precipitation. Precipitation changes can be explained by the changes the total column atmospheric energy budget (O'Gorman et al., 2012). The atmosphere possesses a relatively low heat capacity, and following a perturbation, it rapidly reaches a state where the incoming and outgoing energy fluxes to and from the atmosphere balance each other. In other words, the budget of perturbations between two atmospheric states can be expressed as:

$$L\delta P = \delta R_{Surf} - \delta R_{TOA} + \delta SH = -\delta R_{abs} + \delta SH, \tag{1}$$

where $L$ is the latent heat of condensation, $P$ is precipitation, $R_{TOA}$ and $R_{Surf}$ are the change in the radiative fluxes at the top of the atmosphere and surface, $SH$ is the sensible heat flux change and $\delta R_{abs}$ is the change in absorbed radiation. Niemeier et al. (2013), showed that changes in global latent heat flux dominate changes in sensible heat flux, establishing a roughly linear relationship between precipitation and the discrepancy between the radiative imbalance at the surface and at the top of the atmosphere. Other studies have also shown that changes in precipitation are proportional to the difference between changes in radiation at the surface and in the atmosphere, i.e absorbed radiation (O'Gorman et al., 2012; Kravitz et al., 2013b; Liepert and Previdi, 2009). The atmospheric energy budget can also be utilized to represent precipitation in a transient climate. Given that radiation (and changes in atmospheric absorption) are known to be relatively linearly correlated with global mean precipitation, as evidenced by climate models (e.g (Zelinka et al., 2020)) and observations (Koll and Cronin, 2018) precipitation change can be approximated by a simple equation comprising temperature-dependent and independent components(s):

$$\delta P = a\delta T + F = P_{slow} + P_{fast}, \tag{2}$$

where $\delta T$ is global mean temperature change, $a$ is constant and $F$ are the temperature independent components. Within this equation, $a\delta T$ accounts for all feedbacks attributable to temperature change, including the variation in surface sensible heat flux. This is often referred to as the slow precipitation response or component, which changes over a multi-year timescale

alongside alterations in sea surface temperature. $F$ is referred as fast precipitation response (or component) or rapid adjustment. It can be considered to include the direct radiative forcing, or precisely direct change in absorbed radiation. Thus, at the global scale, a change in global mean precipitation has been shown to be linearly dependent on the absorption part of the induced radiative forcing (Laakso et al., 2020; Myhre et al., 2017; Samset et al., 2016); therefore, a stronger atmospheric absorption of radiation is linked to a decrease in global mean precipitation. Niemeier et al. (2013) investigated the impact of different SRM techniques applied at different altitudes. Their findings show that the precipitation changes predicted by Equation 1 aligns closely with the precipitation changes observed in simulations. Changes in sensible heat flux within their simulations were minimal, suggesting that the calculation of precipitation based on atmospheric absorption is not influenced by the altitude at which the absorption change occurs.

In the case of solar radiation modification generally, the unambiguous impact of this is seen in model simulations, in cases where the GHG-induced radiative imbalance is fully compensated by SRM. Without SRM, there would be an increase in global mean precipitation, driven by a rise in temperature. However, if the temperature increase is offset by SRM, it results in overcompensation and decrease in global mean precipitation (e.g., Kravitz et al. (2013b)). In this case, even though GHG induced radiative imbalance is compensated with SAI, radiative impact of GHG remains in the atmosphere and is still absorbing LW radiation. This is causing a decrease in global mean precipitation even though there is less SW radiation for background atmosphere to be absorbed due to SRM (Laakso et al., 2020). Seeley et al. (2021) studied the idea of concentrating solar dimming at wavelengths, where water vapour has strong absorption bands. This minimized the reduction in the hydrological cycle and simulations showed that it was able to restore mean temperature and precipitation simultaneously. However their study was solely theoretical. The situation is more complicated when aerosols are taken into account. Presumably, there is less SW radiation to be absorbed by background atmosphere under the aerosol layer, but as aerosols are also absorbing LW radiation, it will also slow down the hydrological cycle. Thus, several studies have shown lower global mean precipitation in simulations with SAI compared to an ideal reduction in solar radiation only with similar impacts in global mean temperature (Niemeier et al., 2013; Ferraro et al., 2014). Estimating the total impact of stratospheric aerosols on precipitation is not straightforward, as optical properties of aerosols and impact on SW and LW radiation depend on the size of the aerosols and the injection rate, as described above. In addition, differences in the results between used aerosol modules is expected to translate to large differences in the consequent precipitation responses.

Here, we study how radiative forcings simulated in Part 1 translate to changes in precipitation and temperature. We investigate how aerosol impact on SW and LW radiation changes the atmospheric absorption and further atmospheric energy budget and hydrological cycle. We also study how precipitation changes under different SAI intensities. Furthermore, we examine how these outcomes vary based on the aerosol microphysics model employed to simulate the aerosol fields, as well as the Earth System Model used to simulate climate responses. Simulations are done with three different Earth System Models: EC-Earth, Community Earth System Model (CESM) and Max Planck Institute Earth System Model (MPI-ESM). We implement aerosol optical properties simulated in Part 1 into all three ESMs: thus, stratospheric aerosol optical properties are the same in all three ESMs. In this study, we only consider variations in the strategy in terms of the magnitude of injection rate, but always with the

same injection profile by making injections continuously to the equator only, and ignore changes in strategy using spatial and temporal variation, as done in Part 1.

## 2 Models and Simulations

### 2.1 Models

#### 2.1.1 Earth System Models: EC-Earth, MPI-ESM and CESM

We used three state-of-the-art Earth System Models (ESM), which all include modules for the atmosphere, land and ocean. These models are Max Planck Institute Earth System Model (MPI-ESM1.2) (Mauritsen et al., 2019), Community Earth System Model (CESM2.1.2) (Danabasoglu et al., 2020) and EC-Earth (3.3.1, Döscher et al. (2022)). These models represent a wide range of climate sensitivities (effective climate sensitivity in $CO_2$ quadrupling experiment: MPI-ESM: 3.13 K, EC-Earth: 4.1 K, CESM: 5.15 K) present in CMIP6 models (Zelinka et al., 2020). MPI-ESM consists of the atmospheric models ECHAM6.3, Max Planck Institute Ocean Model (MPIOM) (includes the HAMOCC ocean biogeochemistry model) and the JSBACH land model. CESM 2.0 consists of the Community Atmospheric Model (CAM6),Parallel Ocean Program (POP2) ocean model, the Community Land Model (CLM4), and Community Ice CodE (CICE4) sea ice model. For EC-Earth, atmospheric, ocean, land and ocean biochemistry models are: IFS, NEMO, LPJ-GUESS and PISCES, respectively. Thus these three ESMs do not share the same model components and the results can be considered relatively independent of each other. However radiative transfer module in all three ESMs (and in the aerosol-climate model used to simulate aerosol optical properties of aerosols fields in Part 1) are based on rapid radiative transfer model which uses the 14 SW and 16 LW radiation bands (Döscher et al., 2022; Danabasoglu et al., 2020; Mauritsen et al., 2019). This makes implementation of optical properties of stratospheric aerosols rather straightforward. The resolution of atmosphere used in MPI-ESM, CESM and EC-Earth simulations are T63L47 (1.9° x 1.9°), finite volume 0.9°x1.25° and 32 vertical levels and T255L91 (0.70° x 0.70° ) respectively.

#### 2.1.2 Aerosol-Climate model ECHAM-HAMMOZ used in Part 1

Simulations in Part 1 were done with the aerosol climate model ECHAM-HAMMOZ (ECHAM6.3-HAM2.3-MOZ1.0) (Zhang et al., 2012; Kokkola et al., 2018; Schultz et al., 2018; Tegen et al., 2019). The atmospheric model is the same as in the MPI-ESM version used in this study (Stevens et al., 2013). Simulations were performed with a T63L95 (i.e. 1.9°x1.9°) resolution, which enables resolving the quasi-biennial oscillation (QBO). Aerosols were simulated by two different aerosol modules: the sectional module SALSA, where aerosols are represented by 10 size bins in size space, and the modal module M7, which has 4 modes in size space. 7 years (roughly 3 whole QBO cycles) simulations were done for each scenario. A more detailed description of the model is found in (Laakso et al., 2022).

## 2.2 Implementation of stratospheric aerosol optical properties

Radiation modules in ESMs calculate the impact of aerosols on radiation through three different aerosol radiation properties: i) Aerosol optical depth (AOD) or extinction, which is a quantity that describes how aerosol interacts with the radiation; ii) Single scattering albedo (SSA) which is the ratio of scattering efficiency to total extinction efficiency and the asymmetry factor (ASYM). The aerosol module calculates these quantities at each grid point and radiation wavelength band based on the aerosol size and refractive indices. During the simulations with ECHAM-HAMMOZ in Part 1, AOD, SSA and ASYM were archived as monthly and zonal mean output for 14 SW bands and absorption AOD was archived for 16 LW bands used in the radiation model. In this study, aerosol properties do not vary between years and a 7-year average of radiative properties was taken each month. These aerosol fields were implemented in MPI-ESM, CESM, and EC-Earth by using them as input fields. As mentioned earlier, wavelength bands are the same in all these models and thus, no interpolation to other wavelengths was needed. However, different resolutions were used between models, and thus, aerosols had to be interpolated to corresponding resolutions. Prescribed aerosol properties have two advantages compared to simulating prognostic aerosols in each ESMs: i) ESM simulations without a complex interactive aerosol module are computationally significantly lighter, but nonetheless, this way, the impact of aerosol microphysics is considered, and ii) stratospheric aerosol properties are consistently implemented to all three ESMs.

## 2.3 Quantifying fast and slow responses

The fast temperature-independent response and the slow temperature-dependent feedback response of precipitation and radiation can be quantified using the so-called Regression Method (Richardson et al., 2016). The method can be used for simulations with abrupt/step-change in the climate conditions, e.g., an abrupt change in atmospheric $CO_2$ concentration. By regressing the yearly mean variable of interest (e.g., precipitation) change against the temperature change due to the instantaneous forcing, the fast response is given as the intercept of the regression line and y-axes (dT = 0) and the temperature-dependent feedback response is the slope of the regression line (see Fig. 1). This method is built on the assumption that the variables under analysis are linearly dependent on each other. This is more or less the case e.g., for global mean radiative fluxes and precipitation (Laakso et al., 2022). Thus, this technique is a useful tool to separate the temperature independent quantities as radiative forcing or fast precipitation response as well as effective climate sensitivity and hydrological sensitivity. As shown by Laakso et al. (2020), these quantities can be used to estimate the total global precipitation change in scenarios were SAI is used to mitigate climate change.

## 2.4 Simulated scenarios

### 2.4.1 Stratospheric sulfur injections

In Part 1 we performed several different injection strategies with different injection rates. Here we include only scenarios referred as "baseline" in Part 1 of this study. In this baseline injection scenarios, sulfur was injected continuously as $SO_2$ to

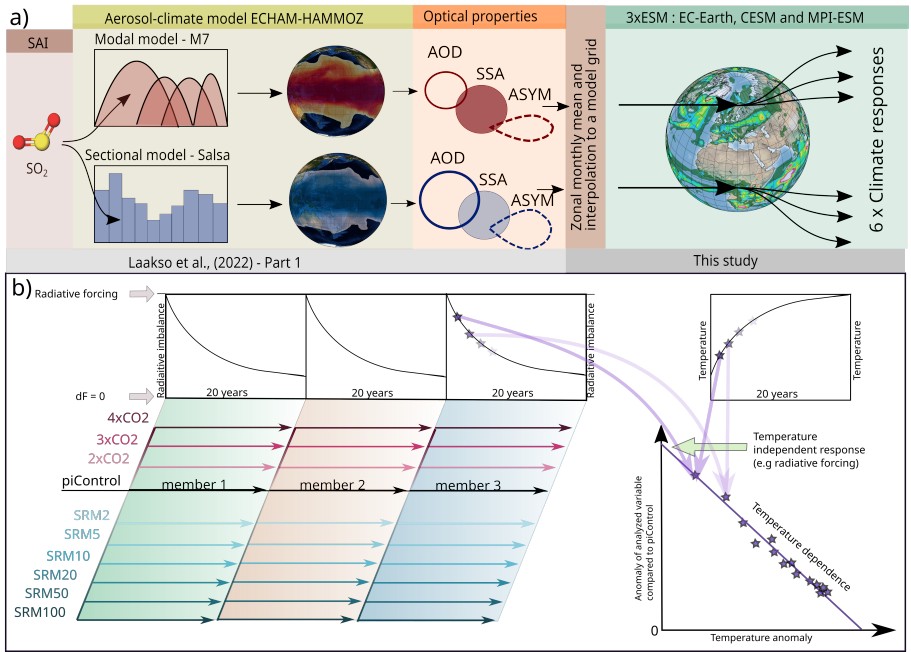

**Figure 1.** a) A schematic of the implementation of aerosol optical properties simulated in (Laakso et al., 2022) to ESMs in this study. b) Simulated regression scenarios and using them to quantify global mean temperature independent responses and quasi-linear dependence on global mean temperature.

a band across all longitudes between the latitudes 10° N and 10° S. The injection was done vertically at 20-22 km altitude. Simulations were done for yearly injection rates of 1, 2, 5, 10, 20, 50, 100 Tg(S)yr$^{-1}$. We excluded here the 1 Tg(S)yr$^{-1}$
simulation as we wanted to concentrate on climatically-relevant, and more signal-to-noise scenarios, to analyse climate impacts in extreme conditions. Simulations were performed with both SALSA and M7 aerosol modules.

### 2.4.2 Regression simulations

The regression simulations with ESMs were started from a preindustrial baseline with GHG and SAI perturbations applied. For SAI, these perturbations were stratospheric aerosol fields (from simulations with 2, 5, 10, 20, 50, 100 Tg(S)yr$^{-1}$ injection
rates) from SALSA and M7 produced in Part 1. In addition, regression simulations with $2xCO_2$, $3xCO_2$ and $4xCO_2$ abrupt forcings were done as well as one simulation in preindustrial conditions without any perturbation. As Richardson et al. (2016) pointed out, a regression length less than 15 years might lead to variation in the quantified fast and feedback responses. On the other hand a longer regression would improve statistics, but then long-scale feedbacks would have a larger role which would lead to a slight nonlinearity. We chose 20 years as the regression length, and to improve the statistics, we simulated three
ensemble members.

**Table 1.** Simulated scenarios

**Aerosol model simulations** - ECHAM-HAMMOZ - Laakso et al 2022

| Scenario | Aerosol model | Injection rate | Injection area | Simulation length |
|---|---|---|---|---|
| **SRM - 2/5/10/20/50/100** | SALSA/M7 | 2/5/10/20/50/100 $Tg(S)yr^{-1}$ $SO_2$ | $10°$ N - $10°$ S, 20-22 km | 7 years |

**Regression simulations** - EC-Earth/CESM/MPI-ESM

| Scenario | Perturbation | Simulation length |
|---|---|---|
| **piControl** | none ($CO_2$ = 280ppm) | 60 yr |
| **2/3/4 x $CO_2$** | $CO_2$: 560/840/1120 ppm | 3 x 20 yr |
| **SRM 2/5/10/20/50/100 - SALSA** | SRM 2/5/10/20/50/100 | 3 x 20 yr |
| **SRM 2/5/10/20/50/100 - M7** | SRM 2/5/10/20/50/100 | 3 x 20 yr |

**Climate equilibrium simulations** - EC-Earth/CESM/MPI-ESM     **Simulation length:** 30+30 yr

| ESM | SALSA aerosols | | | | M7 aerosols | | | | |
|---|---|---|---|---|---|---|---|---|---|
| | SRM2 | SRM5 | SRM10 | SRM20 | SRM2 | SRM5 | SRM10 | SRM20 | SRM50 |
| **EC-Earth** | 382 | 505 | 727 | 1134 | 347 | 400 | 464 | 624 | 1106 |
| **CESM** | 345 | 429 | 576 | 904 | 322 | 348 | 412 | 516 | 904 |
| **MPI-ESM** | 352 | 440 | 556 | 836 | 332 | 358 | 415 | 512 | 782 |

$CO_2$ concentration (ppm) to have presumptive radiative balance with corresponding SAI scenario

### 2.4.3 Radiation equilibrium simulations

In addition to regression simulations, we did simulations where $CO_2$ induced radiative imbalance was compensated by SAI with ESMs. By using regression simulations it was possible to quantify the radiative forcing for each SAI scenarios of different injection rates and 2x$CO_2$, 3x$CO_2$ and 4x$CO_2$ concentration changes in each ESM. As radiative forcing of $CO_2$ depends logarithmically on concentration of $CO_2$, a logaritmic fit can be done for radiative forcings of 2x$CO_2$, 3x$CO_2$ and 4x$CO_2$ concentrations to quantify dependence of radiative forcing on the $CO_2$ concentration. Based on this, in theory we can calculated how large a certain stratospheric sulfur injection rate needs to be to compensate a radiative forcing from a change in $CO_2$ concentration, to maintain radiative balance. Here we define these sulfur injection rate - $CO_2$ concentration pairs to maintain the climate equilibrium and perform simulations with each of the three ESMs. These simulations are 60 years long, and the last 30 years of these simulations were used in the following analysis.

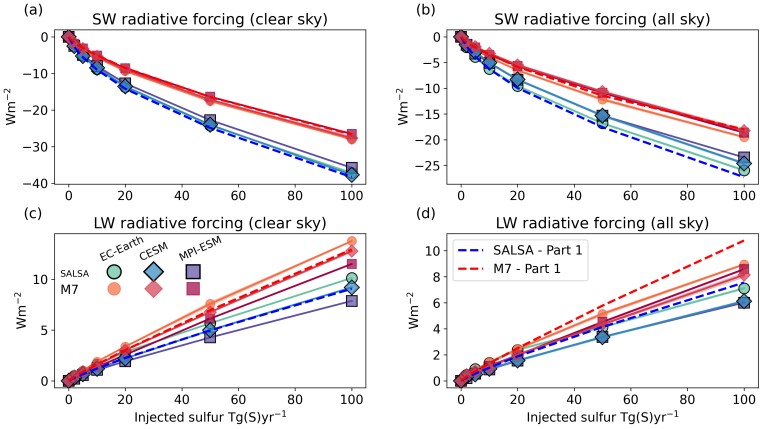

**Figure 2.** Global mean short-wave a) clear sky and b) all-sky and global mean long-wave c) clear sky and d) all-sky radiative forcing as a function of injection rate. Solid lines are radiative forcing from ESM simulations with SALSA and M7 simulated aerosols and based on regression simulations. Dashed lines shows results from Laakso et al. (2022).

## 3 Summary of Part 1 and evaluating the implementation of stratospheric sulfur aerosol fields in ESMs

In Part 1, stratospheric sulfur injections were simulated with a sectional aerosol module SALSA and a modal aerosol module M7. Simulated radiative forcings are shown in Fig. 2. Simulations with both models showed that the SW radiative forcing increased sub-linearly with the injection rate while the increase in LW forcing was more linear. In other respects, there was a significant difference between the model results: SW all-sky radiative forcing was 45-85% higher when based on SALSA simulations than with M7 whereas LW radiative forcing was 32-67% higher in M7 than in SALSA depending on the injection rate. Thus, the total radiative forcing was 88% - 154% higher in SALSA than in M7. Details behind these differences are discussed in Laakso et al. (2022), but generally M7 produced significantly larger aerosols than SALSA. This was caused by both the treatment of the modal size distribution in M7, which prevented aerosols from having an optimal size for scattering under continuous injections, and that in SALSA injected sulfur tended to form new particles instead of condensating on the existing ones, while M7 displayed the opposite behavior.

To ensure that implementation of stratospheric aerosols is done correctly the radiative forcing simulated by each ESMs is compared to the radiative forcing simulated by ECHAM-HAMMOZ in Part 1 (Fig. 2). When comparing these results to the radiative forcings in Part 1, it should be kept in mind that the methods for quantifying radiative forcings was different for ESM simulations compared to Part 1 as the radiative forcing of SAI scenarios in ESM is calculated based on Gregory plots for all-sky SW, LW and total radiative forcings of regression simulations (Gregory et al., 2004). These plots are shown in the Supplement Fig. S1-3. As figures show, global mean radiation flux changes are rather linear with respect to global temperature change. From these figures we can quantify radiative forcings from the y-intercept. In Part 1, radiative forcing was calculated by double radiation call with and without aerosols from simulations with fixed sea surface temperature (SST) and impact of land surface

adjustment of temperature and followed feedbacks are thus included. In addition, the background conditions were different as here the radiative forcings are calculated in preindustrial conditions while in Part 1 simulations were done in year 2005 conditions. Also, radiative properties were not identical between ESM simulations and ECHAM-HAMMOZ simulations as zonal monthly mean of radiative properties of stratospheric aerosols was used in ESM simulations while they were calculated online in ECHAM-HAMMOZ.

Figure 2 shows clear-sky and all-sky SW and LW radiative forcings of SAI as a function of injection rate in MPI-ESM, EC-Earth and CESM. Although the radiative forcings derived from ESMs and simulations in Part 1 are not exactly the same measure, those can be used to see if the implementation of stratospheric aerosols to ESMs was done correctly. The comparison shows the difference in the total radiative forcing between ESMs and Part 1 results, ranging from -27% to 35%. Despite these differences, this comparison provides assurance that the implementation of radiative properties has been carried out correctly especially as radiative forcings between ESMs simulations are in good agreement. Particularly in the case of clear-sky SW forcings, the models exhibit similarities, as expected, due to the similarity in incoming solar radiation, which remains unaltered by clouds. LW radiation and SW all-sky radiation forcings are more dependent on the background conditions and some unique features of each model e.g., clouds, regional distribution of temperature (and emitted LW radiation) as well as ice sheets and surface albedo causing some difference in results between ESMs. In summary, this indicates that the total radiative forcing of SAI can differ slightly among Earth System Models (ESMs), despite having identical radiative properties for the stratospheric aerosols.

## 4 Results

In this section, we begin by employing regression analyses on simulations to estimate the temperature changes in simulated SAI scenarios, based on the effective climate sensitivity. We then proceed to quantify the fast precipitation response and the radiative forcings associated with simulated SAI and $CO_2$ perturbations. These metrics allow us to estimate the extent of $CO_2$ radiative forcing that each simulated SAI scenario could offset. Given the assumption that there should be no change in global mean temperature, the quantified fast precipitation responses can then be utilized to estimate changes in global mean precipitation in scenarios where the radiative forcings of SAI and $CO_2$ are balanced. Lastly, we conduct climate equilibrium simulations for various SAI injection rates and their corresponding $CO_2$ concentrations. These simulations are utilized to examine how estimated precipitation changes, based on the fast precipitation responses, differ from the actual simulated values and to analyze regional responses.

### 4.1 Quantifying fast and slow responses from regression simulations

### 4.1.1 Global mean temperature change under SAI

From regression simulations and regression line for total radiative flux change (Supplement Fig. S3), it is possible to estimate how much global mean temperature will have changed when it does settle down in the new radiative balance after SAI is

started (considering a fixed amount injected per year). This measure is called effective climate sensitivity (the term is generally used for, specifically, the corresponding temperature change for $2xCO_2$ experiment). It does not take into account some of the longer-term nonlinear climate feedbacks that are accounted for in the equilibrium climate sensitivity. Nevertheless, the effective climate sensitivity is a good estimate of temperature changes without simulations spanning over thousands of years,

which would be required to quantify equilibrium climate sensitivity (Gregory et al., 2004). However, it should be kept in mind that temperature change estimates at equilibrium are underestimated if quantified from the effective climate sensitivity; this is especially true here, where we define the slope only from the first 20 years after the induced forcing.

In addition to the magnitude of radiative forcing, the temperature change is influenced by feedback mechanisms, which vary in magnitude for each of the ESM. Some of the simulated scenarios were quite extreme and led to over six degrees change in

global mean temperature already during our 20-year simulation period. This naturally has a large impact on feedbacks, especially the ones that are not always linearly dependent on temperature e.g., cloud and albedo feedbacks. Thus, the dependence of radiative flux change on global mean temperature is not totally linear.

Figure 3a shows the global mean temperature change as a function of the injection rate of SAI in MPI-ESM, CESM and EC-Earth based on effective climate sensitivity. As the figure shows, simulations where SALSA modelled aerosols are implemented

lead to significantly larger global mean cooling compared to M7 aerosols. As expected, larger radiative forcings from SALSA-simulated aerosols translate to a larger global mean temperature change. In addition, based on M7, cooling impact decreases much faster as a function of injection rate than if aerosols are simulated with SALSA, which is also the same pattern in which we saw global mean radiative forcings (Fig. 2). There also are differences in the results between ESMs. Cooling is largest in EC-Earth compared to other ESMs with the same aerosols. In EC-Earth both the total radiative forcing and effective

climate sensitivity parameter were slightly larger (more negative) compared to the other models. Overall, the variation in results between the ESMs was smaller compared to the difference originating from using different aerosol microphysics (M7 vs. SALSA).

In Fig. 3a CESM shows the lowest temperature change in SAI simulations even though based on $4xCO_2$ simulation in (Zelinka et al., 2020), the CESM climate sensitivity was higher compared to EC-Earth and MPI-ESM. This is partly explained

by different responses to the SAI and change in $CO_2$ concentration: In CESM, the climate sensitivity parameter (i.e., the slope of the TOA radiative forcing as a function of temperature change) seems to be lower under $CO_2$ induced warming than when negative radiative forcing was induced with SAI (see Supplement Fig. S3). This means that $CO_2$ induced forcing causes larger temperature change than corresponding forcing induced by SAI. However, based on $4xCO_2$ simulations in this study, the temperature change at the equilibrium based on the regressed line are 7.97 K, 7.80 K and 5.66 K for EC-Earth, CESM

and MPI-ESM respectively, while corresponding values were 8.2 K, 10.3 K and 5.96 K (MPI-ESM-HR) in Zelinka et al. (2020). Note that climate sensitivities reported in Zelinka et al. (2020) are x-intercept values from the Gregory plots for $4xCO_2$ simulations divided by 2. In this study, regression line was fitted based on the first 20 years after induced forcing, whereas Zelinka et al. (2020) quantified it from 150 years. In Supplement Fig. S4 we used Coupled Model Intercomparison Project 6 data from $4xCO_2$ experiment of EC-Earth, CESM and MPI-ESM and showed how the slope of the radiation vs. temperature

change regression line depends on the number of years used to make the fit and to calculate the climate sensitivity. As this

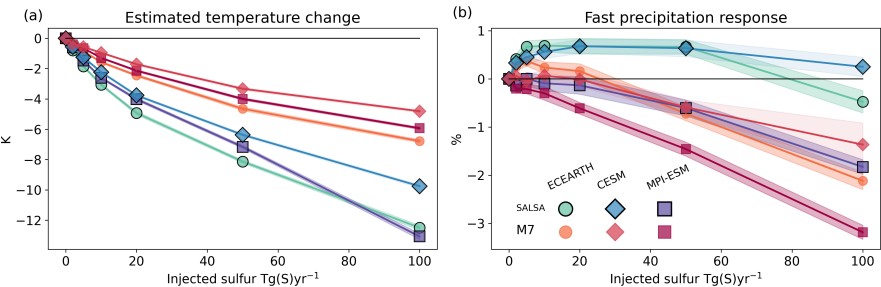

**Figure 3.** a) Estimated temperature changes based on the effective climate sensitivity parameter and b) fast precipitation response as a function of sulfur injection rate based on different models. The shaded area shows the standard error of intercepts of linear fits.

figure shows, the slope becomes smaller and the effective climate sensitivity becomes larger if a larger number of years is used. Magnitude of this change in climate sensitivity parameter is different between models. While in EC-Earth and MPI-ESM the temperature change at the equilibrium quantified from $4xCO_2$ scenario increases from 6.75 K to 8.41 K and 5.78 K to 6.48 K, respectively if 150 years are taken into account instead of 20, in CESM temperature change increases from 8.35 K to 11.71 K. Thus, when the effective climate sensitivity is calculated based on 150-year simulations, the sensitivity appears higher in the CESM model compared to the other two ESMs. However, this difference is not as pronounced when using 20-year simulations.. This characteristic in the CESM results was discussed in Bjordal et al. (2020). It was identified that the increased sensitivity is due to a negative feedback mechanism, which involves a reduction in ice content within clouds in a warming climate. This feedback mechanism becomes less substantial when the climate has warmed sufficiently.

Figure 3a illustrates the divergent temperature change response observed in the MPI-ESM simulation when injected with a rate of 100 Tg(S)yr$^{-1}$ using SALSA-simulated aerosols. This divergence is likely attributed to the non-linear response of shortwave (SW) clouds, as shown in the Supplementary material (see Fig. S5). Notably, this non-linearity becomes apparent when the climate has cooled by over 4 K in the MPI-ESM simulations.

The aforementioned observations emphasize that climate sensitivity is an idealized metric contingent on the timeframe considered. In cases involving SAI compensating $CO_2$ increase, accurately diagnosing the real sensitivity becomes challenging when radiative forcing of increase from $CO_2$ is counteracted with aerosols. As demonstrated later in our analysis, it proves challenging to estimate potential outcomes regarding temperature changes through the simple summation of radiative forcings from both $CO_2$ and SAI. From a broader perspective, arranging climate models in an order based on their level of warming according to a climate sensitivity defined over 150 years appears somewhat arbitrary. This order might change if some other time period to define climate sensitivity is considered.

### 4.1.2 Fast precipitation response under different injection rates

Next, we quantify the fast (temperature-independent) precipitation response under SAI and how this depends on the ESM, aerosol microphysical model used in Part 1 and injection rate. This is an important quantity because it indicates e.g. imperfect

cancellation of LW radiative forcing from $CO_2$. Similarly, as for radiative forcing, the fast precipitation change can be defined
from regression simulations by regressing precipitation against the global mean temperature (see Supplement Fig. S6). Fast
precipitation change is then given by the y-intercept. Fig. 3b shows the fast precipitation response in each simulation as function
of injection rate. For some simulations with small injection rates, the response is small compared to the error bars (shaded area
in the figure). In general, the fast precipitation response is more positive in simulations where SALSA aerosols are used
compared to ones with M7 aerosols in corresponding ESM simulations and the differences between aerosol models become
more pronounced with higher injection rates. These differences among aerosol model results are even more apparent when
the fast precipitation response is presented as a function of radiative forcing. For SALSA aerosols, a lower injection rate can
achieve the same level of radiative forcing as M7, resulting in more significant differences in fast precipitation responses (see
supplement figure Fig. S7a). In addition, the fast precipitation response is non-linear as a function of both injection rate and
radiative forcing. However, the standard deviation of the simulated fast precipitation response between model combinations is
rather linear with respect to the injection rate and the simulated radiative forcing. (see Supplement Fig. S8). This means that
the differences in the simulated fast precipitation response between models become larger with larger injections.

In simulations with SALSA aerosols in EC-Earth and CESM, the fast precipitation is positive for all simulated injection
rates except for 100 Tg(S)yr$^{-1}$ in EC-Earth. With M7 aerosol in EC-Earth and CESM, the fast precipitation response is
slightly positive or small if 20 Tg(S)yr$^{-1}$ or less is injected but negative with 50 or 100 Tg(S)yr$^{-1}$ injection rates. Results
in MPI-ESM differ from CESM and EC-Earth results. In MPI-ESM, the fast precipitation response is small (<0.13% of the
global mean precipitation) with SALSA aerosols with lower than 20 Tg(S)yr$^{-1}$ injection rate. However for larger injection
rates the fast precipitation response was -0.6% and -1.83% lower than the global mean precipitation in piControl simulation.
Fast precipitation response in MPI-ESM-M7 simulations was also much more negative than CESM-M7 and EC-Earth-M7
simulations. Overall quantified fast precipitation response due to the SAI varied between 0.69% increase in global mean
precipitation to -3.19% reduction in precipitation depending on injection rate and ESM-aerosol model combination. Based
on the average hydrological sensitivity in our simulations (Supplement Fig. S6), which were 2.46 %K$^{-1}$ ($\sigma$=0.28 %K$^{-1}$) the
range between the maximum and minimum fast precipitation responses corresponds to a global mean precipitation change
associated with a temperature variation of 1.6 K.

Fast precipitation changes as a function of injection rate can be understood based on the absorbed radiation. As forcing
(change in $CO_2$ concentration or added aerosols) is induced, it changes the radiation absorbed by the atmosphere. E.g. in the
case of higher $CO_2$ concentration, more LW radiation is absorbed. Aerosols also absorb LW radiation, but as aerosols in the
stratosphere reflect solar radiation back to space, there is less radiation to be absorbed by the background atmosphere under
the SAI aerosol layer. Figure 4a shows the net absorption immediately after when forcing is induced (i.e. absorption part of
radiative forcing) versus fast precipitation responses in each simulated scenario. As figure shows, fast precipitation response
and change in absorbed radiation are fairly linearly dependent as shown also by Samset et al. (2016) and Laakso et al. (2020).
This relation was quantified for each model separately even though there are not large differences between models. We can
use this quantity to calculate the individual contribution of SW and LW radiation change to fast precipitation change. This is
shown by dashed and dot-dashed lines in Fig. 4 b,c,d for individual ESM and by using M7 and SALSA aerosols. As less SW

radiation is absorbed, the impact on fast precipitation change is positive whereas increased LW absorption lead to a reduced hydrological cycle. The total impact (calculated based on absorbed radiation) is shown as solid lines in the figure while the markers show actual quantified fast precipitation. As the figure shows, fast precipitation change (markers) and precipitation change, calculated from absorbed radiation, are in good agreement and thus we can be confident that separated examination of absorbed SW and LW radiation can be used to understand the modelled fast precipitation responses.

Regardless of the model used, a common feature of all simulated fast precipitation responses is that the derivative ($dP_{fast}$/Injection rate) of the fast precipitation response as a function of injection rate decreases with larger injection. In other words, the fast precipitation response as a function of injection rate is concave downwards. For some model combinations (EC-Earth-SALSA, CESM-SALSA, EC-Earth-M7), this can be seen as a positive fast precipitation response with a lower injection rate, while fast precipitation change is negative with larger injections. For other model combinations (CESM-M7, MPI-ESM-SALSA, MPI-ESM-M7), the fast precipitation change is negligible or rather small with 2-20 Tg(S)yr$^{-1}$ injection rates, but there is -0.6% - 3.19% reduction in precipitation with larger injection rates. This can be understood in terms of the radiative response seen in absorbed radiation. As mentioned earlier, the impact on SW radiation causes an enhancement of the fast precipitation response, but the SW radiative forcing saturates as a function of injection rate. However, when it comes to longwave (LW) radiation, which exerts a diminishing influence on fast precipitation, the relationship between radiative forcing and the rate of injection tends to be more linear. This means that when considering the net impact of these two components, the significance of the LW radiation impact over the SW radiation becomes larger with a larger injection rate. This turns a positive fast precipitation response into a smaller or negative one, or a negative fast precipitation response even more negative.

The overall precipitation response is influenced by additional factors such as changes in temperature and fast precipitation adjustments caused by other forcing agents. For example, in the case where SAI measures are employed to counterbalance the radiative forcing caused by increased greenhouse gases (GHGs) and compensated warming, there is an observed decrease in the hydrological cycle mainly due to the fast precipitation response of GHG (Laakso et al., 2020). These results show that SAI either reduces or intensifies this decrease depending on the injection rate or models used for simulations. This is studied in the next section.

### 4.1.3  Estimated precipitation change in climate where CO₂- induced radiative forcing is compensated by SAI

In theory, if $CO_2$-induced radiative forcing was compensated by SAI, climate equilibrium should remain and there should not be a change in global mean temperature. This is not completely the case as shown e.g. by Virgin and Fletcher (2022) and as we will see later, but we assume this right now. In this case, where induced radiative forcings are canceling each other out and there is no global mean temperature change, the global mean precipitation change can be calculated by taking a sum of the fast precipitation responses of induced forcings (Laakso et al., 2020). In this case, this equates to the sum of the fast precipitation changes caused by SAI and those by $CO_2$ concentration increases.

First, we need to calculate the extent to which each simulated SAI experiment can compensate for changes in $CO_2$ concentration. We conducted four regression simulations with varying $CO_2$ concentrations: preindustrial, 2x$CO_2$, 3x$CO_2$, and 4x$CO_2$. Using these simulations, we calculated the radiative forcing for each scenario. Since we know that the radiative forcing induced

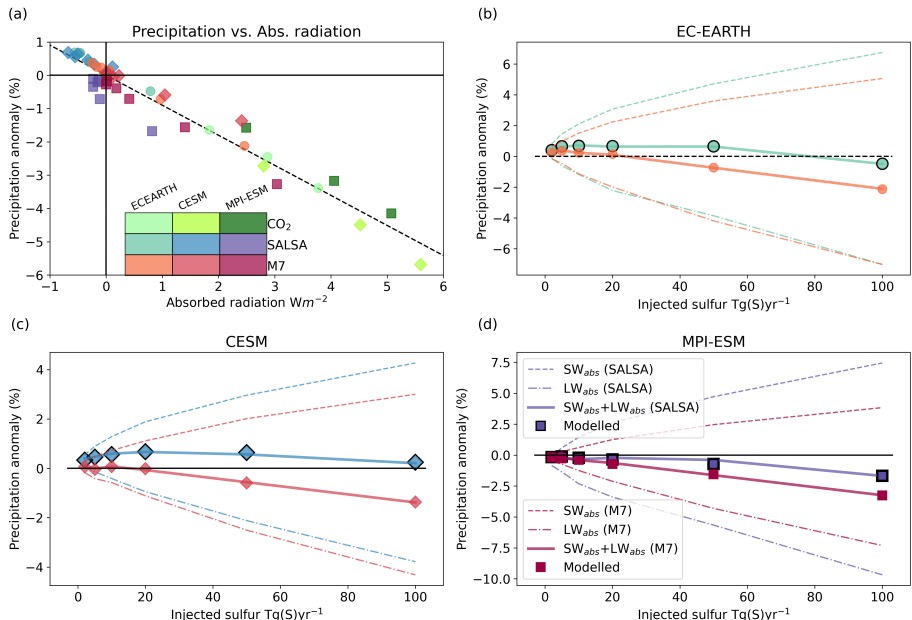

**Figure 4.** a) Regression of fast precipitation response versus atmospheric absorption, b-d) precipitation anomaly as a function of injection rate in EC-Earth, CESM and MPI-ESM respectively. Markers are quantified from regression simulations by regressing precipitation against temperature while lines are calculated from atmospheric absorption based on the relation in a). The dashed line is precipitation change caused by SW absorption, the dash-dotted line is based on LW absorption and the solid line is the sum of these SW and LW components whereas markers are modelled fast precipitation responses from regression simulations.

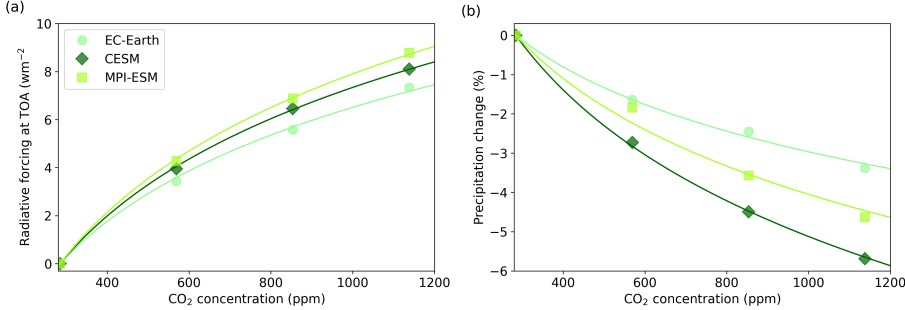

**Figure 5.** a) Radiative forcing at the top of the atmosphere and b) fast precipitation response as a function of atmospheric $CO_2$ concentration based on logarithmical fit for results from piControl, 2x$CO_2$, 3x$CO_2$ and 4x$CO_2$ scenarios

by $CO_2$ depends logarithmically on the atmospheric concentration of $CO_2$, we used a logarithmic fit to determine the radiative forcing for each of the four simulated values (see Fig. 5). This function provides the radiative forcing for a particular $CO_2$ concentration for each of the three ESMs. By utilizing this function and the radiative forcings for each SAI simulation, we can

determine the specific $CO_2$ concentrations for each SAI experiment at which there would be a climate equilibrium. Table 1 and Fig. 6a display these $CO_2$ concentrations.

We can now also reverse the aforementioned question and ask how large a sulfur injection is required to be to offset the radiative forcing resulting from a certain increase in atmospheric $CO_2$ concentration. This is shown in Fig. 6a based on different model combinations (ESM-aerosol model). The total range of the estimated amount of required sulfur injection rates between model combinations is large. The greatest discrepancies are between simulations that utilize SALSA aerosols versus those that use M7-simulated aerosol properties. As expected, due to the lower radiative forcing produced by aerosols simulated by M7, significantly higher injection rates are needed to compensate for certain $CO_2$-induced forcing, compared to simulations that use SALSA aerosols. There are some differences between ESMs when the same aerosols were used. Regardless of the aerosols used and considered $CO_2$ concentration, the estimated injection rates were similar between CESM and MPI-ESM simulations. However, in the case of EC-Earth, notably, less sulfur was needed. For instance, to offset the radiative forcing of an 800ppm $CO_2$ concentration, EC-Earth simulations necessitated 30-40% less sulfur annually compared to the corresponding CESM and MPI-ESM simulations.

To estimate the potential global mean precipitation changes in the previously mentioned scenarios, we assume a radiative balance that does not cause any temperature change. Similar to $CO_2$-induced radiative forcing, the fast precipitation response as a function of $CO_2$ concentration is calculated through a logarithmic fit for preindustrial, 2x$CO_2$, 3x$CO_2$, and 4x$CO_2$ fast precipitation responses (see Fig. 5b). Assuming that there is no temperature change if the radiative forcing from $CO_2$ and SAI are offsetting each other, we can calculate the global mean precipitation change as the sum of the fast precipitation response of $CO_2$ from the fitted logarithmic function and the corresponding SAI experiment. Figure 6b displays the resulting global mean precipitation changes.

Estimates of precipitation changes depend substantially on different model combinations. As illustrated by Fig. 5b and Supplementary Fig. S6, the fast precipitation response to a quadrupling of $CO_2$ levels varied significantly, ranging from a decrease of 3.38% in the EC-Earth simulations to an increase of 5.6% in the CESM simulations. However, the fast precipitation response to SAI accounts for differences of up to 3.5% in global mean precipitation, as illustrated by the fast precipitation response component in Supplementary Figure S7b. If M7 aerosols are used, models project a greater reduction in precipitation relative to the baseline than if SALSA aerosols are used with the same ESM. This is expected because the absorption of LW radiation is greater in M7-based aerosols than in SALSA, and more sulfur is required to offset $CO_2$-induced forcing when using M7 aerosols. There is not a large difference in precipitation changes between MPI-ESM and CESM when using both M7 and SALSA aerosols, while in EC-Earth, there is a smaller decrease in precipitation compared to simulations with two other ESMs regardless of the aerosol model used. This is due to two factors in EC-Earth simulations: the smaller magnitude of the fast precipitation response to $CO_2$ (as shown in Fig. 5b) compared to MPI-ESM and CESM, and the more positive fast precipitation response to SAI when the injection rate is adjusted to match the radiative forcing of $CO_2$ (refer to Fig. S7b).

However, as indicated by Supplement Figure S6, employing a simplistic approach using fast and slow responses to estimate precipitation changes may not be straightforward. Supplement Figure S6 reveals variations in the hydrological responses among three Earth System Models (ESMs), particularly in the variation of the hydrological sensitivity (i.e., the slope in the figure)

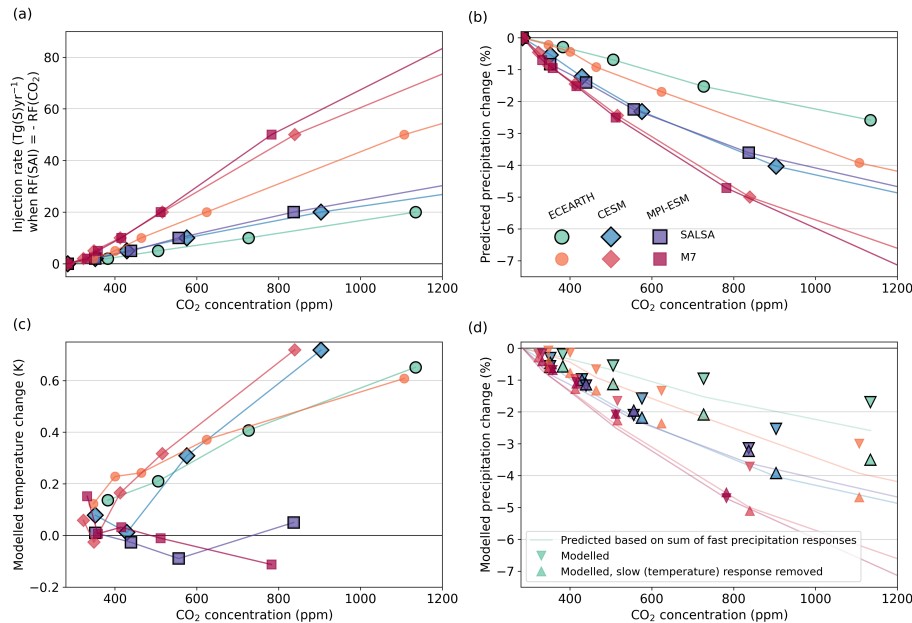

**Figure 6.** a) The estimated injection rate of stratospheric sulfur injections and b) estimated precipitation change in different model combinations if radiative forcing (RF) of $CO_2$ concentration is compensated by SAI. Global mean precipitation change is calculated as the sum of the fast precipitation changes from SAI and $CO_2$, assuming that there is no change in global mean temperature. Based on the logarithmic relationship between radiative forcing and fast precipitation response to $CO_2$ concentration (as shown in Fig. 5), the CO2 concentration and the subsequent fast precipitation response can be determined from the logarithmic fit so that the radiative forcing aligns with the simulated radiative forcing for SAI. c) Simulated changes in a) global mean temperature and b) precipitation under SAI - $CO_2$ pair scenarios (as illustrated in a), assuming a state of climate equilibrium. In d) "triangle down"-marker shows actual simulated precipitation, "triangle up"-marker shows adjusted values based on hydrological sensitivity and assuming zero global mean temperature change and solid line shows estimated precipitation change based on fast precipitation changes (same as in b)).

across various simulated forcing agents. Simulations using CESM and MPI-ESM suggest that the hydrological sensitivity increases with larger injections, but the range of this increase differs significantly from the sensitivity observed in simulations where $CO_2$ concentration was perturbed. Conversely, in EC-EARTH simulations, hydrological sensitivity ranged from 2.39 to 2.48 $\%K^{-1}$ in scenarios with $CO_2$ perturbations, while in SAI scenarios, the total range was 2.79 - 3.22 $\%K^{-1}$. This discrepancy is a crucial factor to consider, especially in cases where the forcing induced by $CO_2$ and SAI does not fully offset each other but might also have an impact when those are expected to compensate each other.

## 4.2 Results of climate equilibrium simulations

Next, we conducted simulations where radiative forcing from $CO_2$ and SAI compensated each other based on the SAI experiment-atmospheric $CO_2$ concentration pairs calculated in the previous section. These simulations allowed us to observe how well our estimations for precipitation changes (Fig. 6b) held and, additionally, to simulate regional changes. The simu-

lations were conducted under pre-industrial conditions with prescribed SAI aerosol fields and changing $CO_2$ concentrations to corresponding levels to maintain climate equilibrium. Another option would have been to simulate specific $CO_2$ concentrations and scale the aerosol optical properties to match the radiative forcing of $CO_2$. However, due to the non-linear relationship between aerosol size distribution and optical properties of stratospheric aerosols in response to injection rates, scaling would have yielded slightly divergent outcomes compared to simulations in which intermediate injection rates were simulated using the aerosol models. Moreover, adopting the scaling approach would have resulted in the loss of specific characteristics unique to both the M7 and SALSA aerosol models.

These climate equilibrium simulations were only conducted for cases where the $CO_2$ concentration was below 1200 ppm. As a result, the maximum injection rate that was simulated was 20 Tg(S)yr$^{-1}$ using SALSA aerosols and 50 Tg(S)yr$^{-1}$ using M7 aerosols. These simulations were 60 years long, and the last 30 years were used in the analysis.

### 4.2.1 Global mean temperature change in climate equilibrium simulations

Figure 6c shows global mean temperature changes in the climate equilibrium simulations as a function of atmospheric $CO_2$ concentration. Note that these simulations now include SAI aerosol fields and change atmospheric $CO_2$ concentration which is specific for injection rates and the model (seen Table 1 and Fig. 6a). From Fig. 6c, it is clear that the assumption of having no global mean temperature change holds only for MPI-ESM, while in EC-Earth and CESM simulations, there is global mean warming up to 0.72 K in scenario with 20 Tg(S)yr$^{-1}$ injection rate and 904 ppm $CO_2$ concentration with CESM-SALSA model combination. Based on Fig. 6c, the primary factor influencing the extent of remaining warming is the ESM, whereas the influence of the aerosol model has a comparatively minor impact. Although in simulations conducted with CESM and EC-Earth models, where there is observable residual warming, the magnitude of this residual warming tends to be greater when the M7 aerosol optical properties are employed across most of the simulations. Nonetheless, the warming observed in these simulations underscores the presence of nonlinearity when combining individual experiments involving $CO_2$ and SAI. This is also seen in earlier studies, although those studies primarily focused on reducing the solar constant and the approach to provide initial guess for solar constant reduction (Virgin and Fletcher, 2022; Russotto and Ackerman, 2018). When applying the Gregory method to assess radiative forcings in these scenarios, the calculated values range 0.17-1.25 W/m2 in CESM and EC-Earth simulations and is up to -0.24 W/m2 in MPI-ESM simulations which aligns well with the simulated temperature (see Supplement, Fig. S9). However, as Supplement Fig. S9 shows, the Gregory method does not work well in this case, at least for EC-Earth and CESM, where the slope defined for individual simulations varies, and there is no clear linear dependence between the total radiative flux change and global mean temperature. In general, the presence of nonlinearities when combining $CO_2$ and SAI is expected based on the findings presented in Supplementary Fig. S3. The figure shows that the slopes of the fitted lines (i.e $\delta T/\delta F$) on the Gregory plots are, on average, 41% lower in EC-Earth simulations and 27% lower in CESM simulations for $CO_2$ experiments compared to SAI experiments. However, in the case of MPI-ESM simulations, there was no significant difference. We will discuss possible physical reasons for the residual global mean warming in CESM and EC-Earth simulations in section 4.2.3

#### 4.2.2 Simulated global mean precipitation change in climate equilibrium simulations

As there is an increase in global mean temperature in these simulations, the actual simulated global mean precipitation differs fundamentally from the estimated ones in the previous section. In Fig. 6d, the solid lines show the estimated precipitation change using fast precipitation responses (same as Fig. 6b). Triangle markers that point down show the actual simulated precipitation in these scenarios. As global mean temperature changes were rather small in the MPI-ESM simulations, the actual simulated precipitation changes were close to the estimated ones. However, as there was a slight warming in the EC-Earth and CESM simulations, global mean precipitation is more positive than those estimated from the sum of fast precipitation responses. Hydrological sensitivity (i.e., the ratio of precipitation change to temperature change) can be used to remove the impact of global mean temperature on precipitation. Triangle markers that point up are adjusted values of simulated precipitation by counteracting the impact of temperature change. Now, the adjusted values from CESM simulations are close to the estimated ones. For EC-Earth, this adjusement corrects precipitation values to the direction of estimated ones, but it over-adjusts them for most of the simulated scenarios. It remains unclear why this temperature adjustment leads to an overestimation in the results for EC-Earth simulations. However, this could be related to the larger hydrological sensitivities for SAI compared to $CO_2$ perturbations, as discussed in section 4.1.3. Although there are discrepancies between the actual simulated values and the estimated ones, this analysis shows that estimating the total precipitation change based on the sum of fast precipitation responses of SAI and change in $CO_2$ concentration gives rather good results even though there are some changes in global mean temperature. The main conclusions also hold after analyzing the actual simulations: there are large variations in global mean precipitation between models, and larger aerosols based on M7 lead to a larger reduction in precipitation than those simulated by SALSA.

#### 4.2.3 Regional temperature responses in the equilibrium scenarios

Figure 7 shows the zonal mean, and Fig. 8 the regional temperature response in the climate equilibrium simulations for selected scenarios. The regional responses for all scenarios are shown in supplementary Fig. S10-11. Several earlier studies have shown that compensating GHG-induced warming with low-latitude SAI or SRM generally leads to residual warming in high latitudes and overcooling at low latitudes(e.g., Schmidt et al. (2012); Kravitz et al. (2013a); Visioni et al. (2021)), unless the injections are explicitly targeted to avoid this imbalance ((Kravitz et al., 2017, 2019; MacMartin et al., 2017)). Laakso et al. (2022) demonstrated that the radiative forcing from SAI is primarily concentrated around the Equator for aerosols simulated using both SALSA and M7 models. There was also significant clear-sky zonal forcing observed at the latitudes of 50°N and 50°S. However, the presence of clouds in these regions reduced the aerosol all-sky radiative forcing. Aerosol optical properties were consistently applied across all three ESMs, but variations in cloud cover and properties among the ESMs can lead to differences in the actual radiative impact of aerosols.

Here overcooling of tropics is seen only in MPI-ESM simulations. As there was global mean warming in the CESM and EC-Earth simulations, there are fewer and smaller regions compared to the MPI-ESM simulations that show negative temperature anomaly, especially within scenarios with larger atmospheric $CO_2$ concentrations and SAI. However, in simulations using these

models, the temperature gradient between low and high latitudes changes in a manner similar to that in MPI-ESM and there are large areas where temperature change is not statistically significant, even in the higher injection scenarios. There are also some differences in temperature patterns in EC-Earth and CESM simulations: Arctic warming is slightly stronger in EC-Earth simulations compared to the results of the two other models, while CESM simulations show stronger warming over the tropical ocean and especially in stratocumulus regions. Overall, temperature changes (residual warming at high latitudes for all models and overcooling at low latitudes in MPI-ESM) are amplified with larger atmospheric $CO_2$ concentrations and injection rates of SAI. There is no clear distinguishable difference in regional patterns caused by the aerosol model, i.e., SALSA versus M7 aerosols. However, estimating the impact of aerosols is not straightforward as atmospheric $CO_2$ concentration is adjusted so that its global mean radiative forcing compensates for the radiative forcing of SAI. Thus atmospheric $CO_2$ concentration is not the same in climate equilibrium simulations between M7 and SALSA-simulated aerosol for the same injection rate.

Regional temperature patterns provide some hints about the reason for the global mean residual warming observed in EC-Earth and CESM simulations. Although the global mean radiative forcing of SAI compensates for the global mean radiative forcing from increased $CO_2$ concentration based on single forcing experiments, the zonal impacts are not uniform (see the mean total radiative flux change in the first 5 years of SRM20-SALSA simulations in Fig. S12 of the Supplement). The gradient of solar radiation between high and low latitudes is steeper than for thermal radiation. Thus, the combination of a uniform reduction in both incoming SW and outgoing LW radiation leads to fundamentally less radiation in lower latitudes and more radiation at high latitudes, even though they would compensate each other on average. Furthermore, concerning stratospheric aerosols, the impact on radiative forcing is more pronounced at the Equator and latitudes around 50 degrees north and south where aerosol concentration is large due to the atmospheric circulation (Laakso et al., 2017). Thus radiative forcing is larger compared to the latitudes in between these regions (i.e 20-30°N and S latitudes) and it is particularly prominent when contrasted with the impact on higher latitudes (Laakso et al., 2022). Thus it is possible that when high latitudes warm up, ice and snow start to melt, and less solar radiation is reflected back into space.

On the other hand, net radiative flux changes seen in Fig. S10 are more negative over Tropics in MPI-ESM simulations compared to two other models. Additionally, for example, the warmer pattern in MPI-ESM simulations over the North Pacific and cooler over Alaska indicates that less warm air is transferred to Alaska and the Arctic by the North Pacific current, which might prevent the melting of Arctic sea ice. Similar patterns are observed in those CESM simulations where global mean warming was small (e.g., SRM2-SALSA/M7, SRM10-M7). CESM simulations show warming in stratocumulus areas which indicates changes in cloud radiative forcing. This is supported by Fig. S13 of Supplement, which illustrates the change in SW radiation fluxes between simulation with SAI together with increased $CO_2$ concentrations compared to the piControl simulation. As we can see, there is a reduction in reflected SW radiation over stratocumulus cloud areas in CESM simulations even though generally more radiation is reflected due to the SAI. Similar cloud adjustments have been noted in previous studies, where simulations have been conducted to explore the linearity of responses to SAI and $CO_2$ responses are simulated (Virgin and Fletcher, 2022). CESM simulations with larger $CO_2$ concentration and large SAI injection rate (e.g SRM20-SALSA and SRM50-M7 (supplement Fig S10.)) are showing cooling in the North Atlantic which is associated to the weakening of the

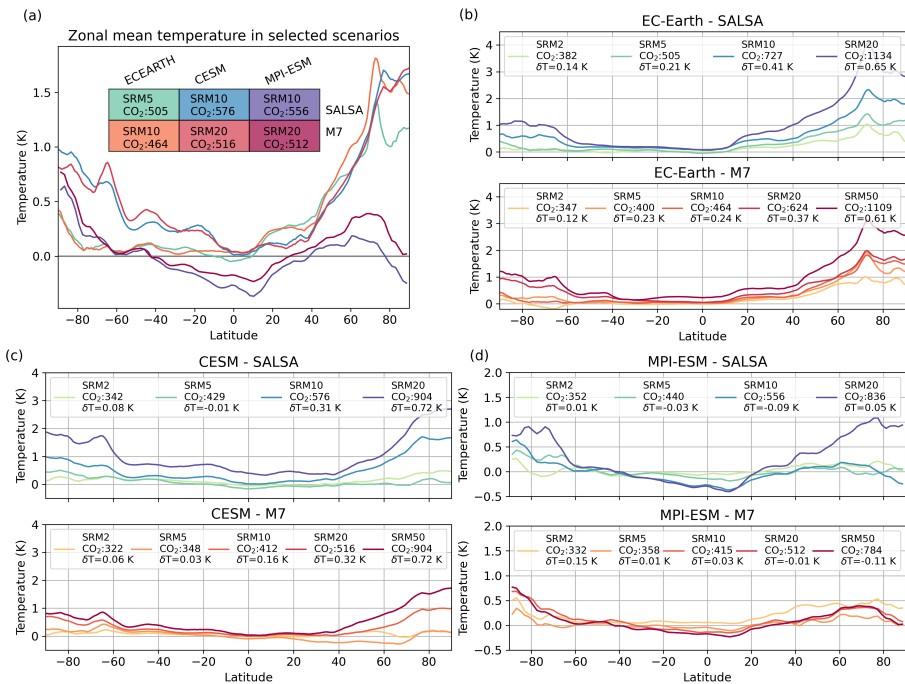

**Figure 7.** Zonal mean temperature (a) for climate equilibrium scenarios where atmospheric $CO_2$ concentration was between 464-576 ppm and climate equilibrium scenarios for (b) EC-Earth, (c) CESM, and (d) MPI-ESM. In these simulations, the $CO_2$ concentration was adjusted to counterbalance the radiative forcing from a specific injection rate, as determined by regression simulations. $\delta T$ in the legends shows residual global mean temperature.

Atlantic Meridional Overturning Circulation seen also in simulations with global warming (Meehl et al., 2020; Fasullo and Richter, 2023).

### 4.2.4 Regional precipitation responses in the equilibrium scenarios

Figure 9 shows differences in zonal mean precipitation and Fig. 10 shows the regional precipitation difference between climate equilibrium simulation and piControl as an average of 30 years. In earlier sections, we focused on yearly mean values or mean periods over several years. Therefore, in this section, we focus only on averages over the years and do not analyze seasonal impacts. The regional precipitation change patterns show a shift of Intertropical Convergence Zone (ITCZ) and a reduction of precipitation over land, but as for the temperature impacts, the changes intensify when the $CO_2$ concentration in the atmosphere and the injection rate of SAI increase. Statistical significant precipitation changes are observed only in a minority of regions, particularly in weaker-forcing cases with low $CO_2$ increase and SAI injection rates. Similar to the temperature changes discussed in the previous section, there are no significant differences in regional patterns of precipitation change between using M7 or SALSA aerosols. However, larger $CO_2$ concentration and larger forcing from SAI when SALSA aerosols are used lead to a more intensive impact than the corresponding injection rate using M7 aerosols.

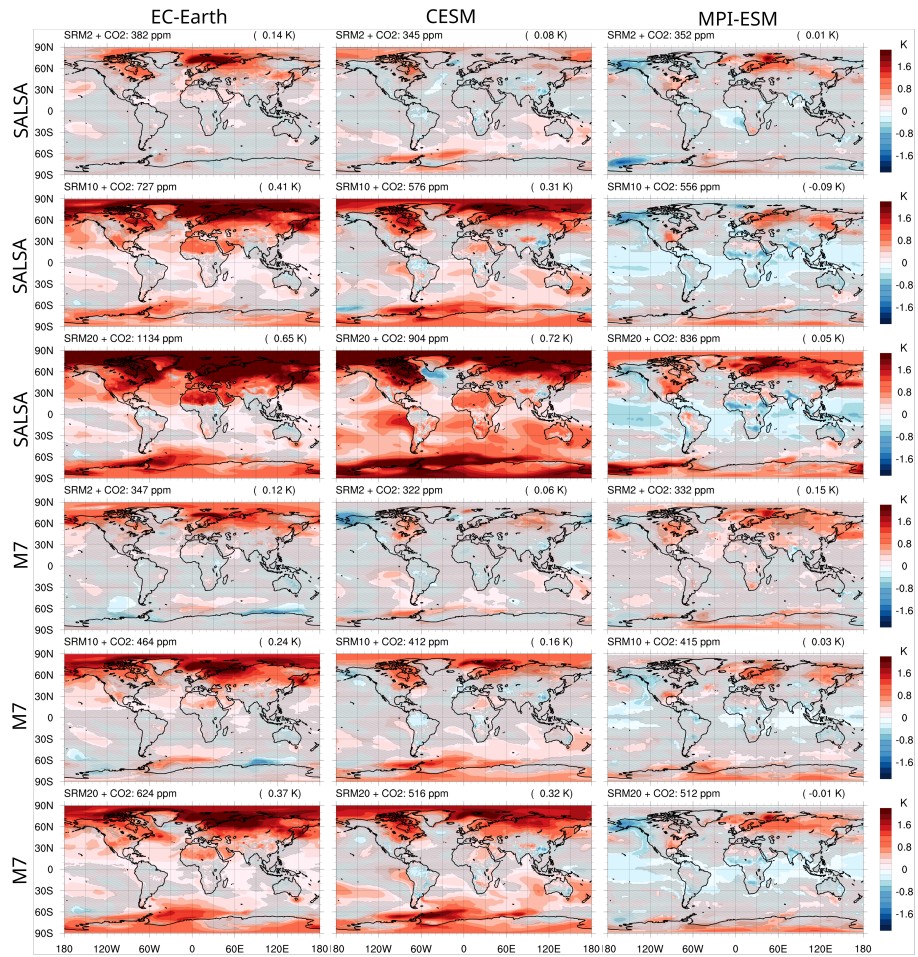

**Figure 8.** Differences in regional temperature patterns between the climate equilibrium scenarios and piControl scenario. EC-Earth results are in the left column, CESM results are in the middle and MPI-ESM results in the right column. The number inside the parentheses in the upper right corner of each panel is a global mean temperature change in each scenario. Hatching indicates regions where the temperature change is not statistically significant based on the Wilcoxon signed-rank test (p-value < 0.05) (Wilcoxon, 1945).

There are some differences in regional patterns of precipitation between ESMs. The differences between model responses
are particularly noticeable when $CO_2$ concentration and SAI forcing are high. As depicted in Fig. 9, there is a decrease in precipitation over the Equator in the EC-Earth and MPI-ESM simulations, which indicates a broadening of the ITCZ. Conversely, in the CESM simulations, the ITCZ is observed to be shifting southward. EC-Earth results show mostly statistically significant increases in precipitation, whereas this is not the case in MPI-ESM and CESM results, despite strong warming over the Arctic area in CESM. EC-Earth and MPI-ESM results indicate a relatively large increase in precipitation over the Sahel
region, while CESM results show mostly statistically insignificant changes. According to CESM and MPI-ESM results, the tropical region in Central Africa is receiving less precipitation, while EC-Earth shows a much smaller precipitation change in

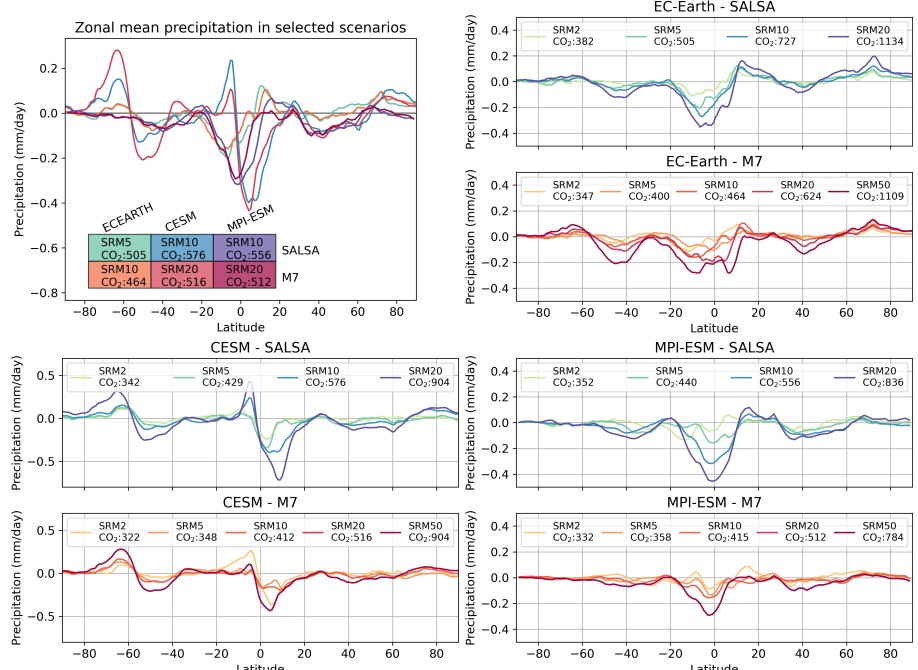

**Figure 9.** Zonal mean precipitation (a) for climate equilibrium scenarios where atmospheric $CO_2$ concentration were between 464-576 ppm and climate equilibrium scenarios for (b) EC-Earth, (c) CESM and (d) MPI-ESM.

that region. CESM results indicate a strong intensification of precipitation over the Equator, which is not observed in EC-Earth and MPI-ESM results. On the other hand, there are also regions where model results agree with each other. Generally, precipitation decreases over oceans (except for the Equator in CESM results). Precipitation increases over Australia, as well as in the

555 Arabian Peninsula, Pakistan, and India, but decreases over the northern parts of South America.

## 5 Conclusions and discussion

In Laakso et al. (2022), we simulated SAI of different magnitudes using the sectional (SALSA) and modal (M7) aerosol schemes, which showed significant differences in the simulated radiative forcings between the two aerosol models. In this study, we implemented the simulated radiative properties into three ESMs (EC-Earth, CESM, and MPI-ESM) to study the

560 temperature and precipitation responses under different magnitudes of SAI, based on the results from the two aerosol schemes. This was done through two sets of simulations, using the aerosol optical properties from the preceding SALSA and M7 simulations for injection rates of 2-100 Tg(S)yr$^{-1}$: 1) regression simulations were conducted under preindustrial conditions with the additional instantaneous forcing, and 2) alleged climate equilibrium simulations were performed, where the global mean radiative forcings of $CO_2$ increase and SAI compensated each other.

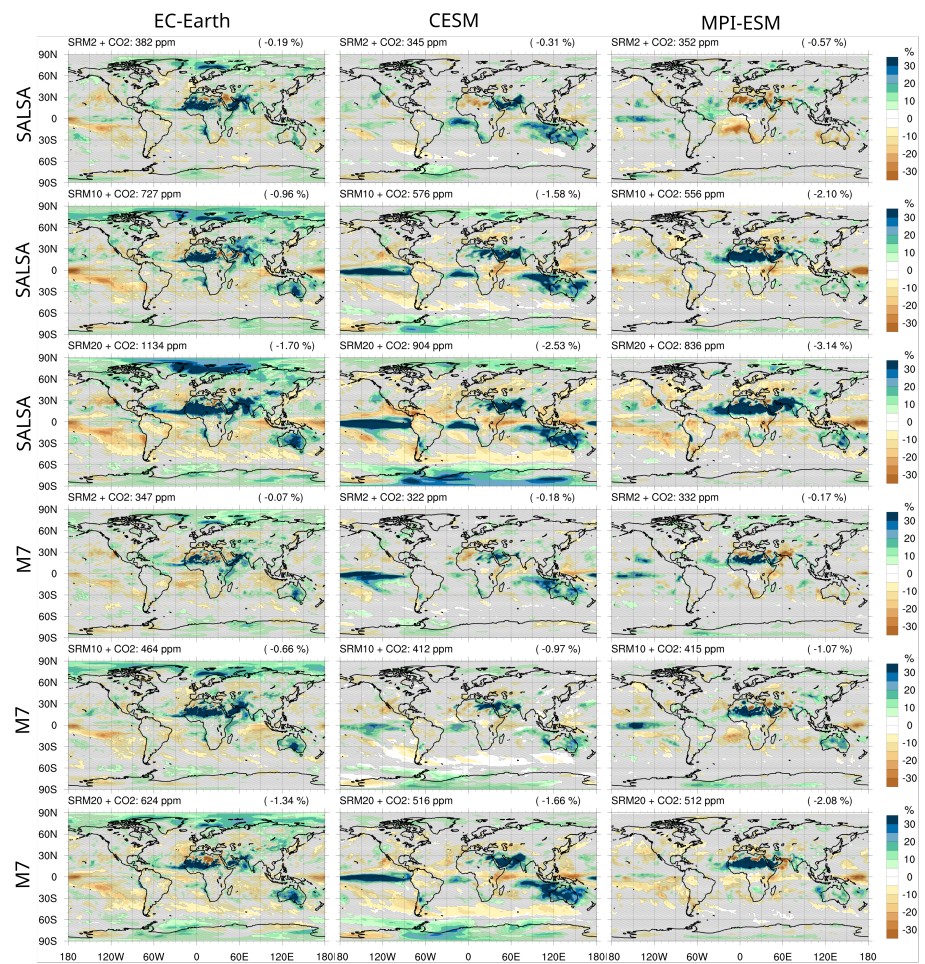

**Figure 10.** Differences in regional precipitation patterns between the climate equilibrium scenarios and piControl scenario. EC-Earth results are in the left column, CESM results are in the middle and MPI-ESM results are in the right column. The number inside the parentheses in the upper right corner of each panel is a global mean relative precipitation change in each scenario. Hatching indicates regions where the precipitation change is not statistically significant based on the Wilcoxon signed-rank test (p-value < 0.05).

There was a significant difference in responses between models. For example, the radiative forcing of SAI, with an injection rate of 20 Tg(S)yr$^{-1}$ varied between -3.48 Wm$^{-2}$ and -7.16 Wm$^{-2}$, depending on the ESM-aerosol model combination (Fig. 2). Based on the significant differences in radiative forcings outlined already in Part 1, most of the variations in radiative forcings among ESM-aerosol model combinations can be attributed to differences in the SALSA and M7 aerosol simulations. In these simulations in this study, an injection rate of 20 Tg(S)yr$^{-1}$ resulted in radiative forcings ranging from -6.75 Wm$^{-2}$ to

570  -7.16 Wm$^{-2}$ using the SALSA-simulated aerosols. In contrast, with M7 aerosols, the radiative forcing ranged between -3.48 Wm$^{-2}$ and -4.07 Wm$^{-2}$. However, an even greater variation in results was observed when examining how these differences in radiative forcings translated into climate impacts. Based on the climate sensitivity parameter from the first 20 years of

our regression simulations for 20 Tg(S)yr$^{-1}$ injection rate, the projected range for global mean temperature change spanned between -2.2 K and -5.2 K. Further, this range can be subdivided into two groups: -(2.2 - 2.8 K) for ESM simulations where M7 aerosols were used, and -(4.0 - 5.2 K) for simulations based on SALSA. Simulated temperature change was smallest in CESM simulations based on both SALSA and M7 aerosols despite that climate sensitivity of CESM has been shown to be markedly higher compared to other two ESMs (Zelinka et al., 2020). This discrepancy was attributed to determining the climate sensitivity parameter based on a 20-year span rather than a 150-year period, as e.g., in Zelinka et al. (2020), but also differences in responses to the SW vs. LW radiative forcing or cooling vs. warming. Except for most extreme impact simulated in this study (simulating 100 Tg(S)$^{-1}$ injection rate with SALSA), the temperature change was largest in EC-Earth simulations. This resulted from both a slightly larger climate sensitivity parameter (based on a 20-year span) and a larger simulated radiative forcing of SAI in EC-Earth compared to the other two ESMs. Overall, drawing from these results and a comparison with the climate sensitivities reported in (Zelinka et al., 2020), it should be kept in mind that effective climate sensitivity is not a straightforward parameter. Its interpretation is complicated by its sensitivity to external factors, such as the type of forcing agent (which affects shortwave vs. longwave radiation) and the length of the simulation period (e.g., 20 years vs. 100 years). Moreover, sensitivity to these external factors varies across different models..

Based on the radiative forcings quantified from regression simulations we estimated the annual sulfur injection required to compensate for the radiative forcing of $CO_2$ ranging from preindustrial concentration to 1200ppm (see section 4.1.3). The results varied significantly among different combinations of ESM-aerosol models. By making interpolation between simulated results, offsetting the radiative forcing from 500 ppm atmospheric $CO_2$ concentrations, required sulfur injection rate varied between 5-19 Tg(S)yr$^{-1}$ between aerosol-ESM model combinations. As expected, the most significant differences arose from the choice of aerosol model used for simulations. Estimates for the required injection rate varied from 5 to 8 Tg(S)yr$^{-1}$ when SALSA aerosols were employed, and from 12 to 19 Tg(S)yr$^{-1}$ with M7-simulated aerosols. By using quantified fast precipitation responses, we were able to estimate subsequent changes in global mean precipitation under these scenarios, assuming no alteration in global mean temperature due to the presumed climate equilibrium. This led to a reduction in precipitation across all simulated scenarios. In the aforementioned 500ppm atmospheric $CO_2$ concentration and SAI scenario the resulting reduction compared to preindustrial climate in global mean precipitation ranged from 0.7% to 2.4% between different model combinations (Fig. 6). In the same $CO_2$ concentration within the same ESM, a larger decrease in precipitation was consistently observed when M7 aerosols were used compared to SALSA aerosols. However, when considering different ESMs, there was no distinct separation between SALSA aerosol-based simulations, which exhibited a global mean precipitation reduction ranging from 0.7% to 1.8%, and M7-based simulations, which showed a reduction ranging from 1.4% to 2.4%." When conducting the actual simulations for these presumed climate equilibrium scenarios, we observed that the assumption of no change in global mean temperature was valid only for the MPI-ESM simulation. In contrast, in the CESM and EC-Earth simulations, there was global mean warming of up to 0.7 K in certain runs. Hence, the range for simulated precipitation reduction in the presumed climate equilibrium scenario for 500 ppm was 0.5% to 2.0%, which was slightly different from the earlier estimate.

We looked deeper into global precipitation impacts caused directly by SAI analyzing its fast precipitation response. There were large differences between fast precipitation responses between model combinations: the CESM-SALSA combination

simulated positive fast precipitation changes from 0.25 to 0.85% increase in global mean precipitation with injection rate levels of 2-100 Tg(S)yr$^{-1}$, while the fast precipitation response was -(0.2 - 3.19%) in MPI-ESM-M7 (Fig. 3). However, two systematic patterns emerged in the results: 1) precipitation was always more negative in simulations where M7 aerosols were used compared to SALSA aerosols with the corresponding injection rate 2) All simulations with each model combination showed that the slope of the fast precipitation function with respect to injection rate decreases with a larger injection rate. In other words, the results of all models indicate that the positive fast precipitation response turns negative if the injection rate is increased enough, and negative precipitation change intensifies to an even greater extent when the injection rate is increased.

The fast precipitation responses can be understood based on SAI impact on atmospheric absorption. As the global mean fast precipitation response is negatively correlated with global mean absorbed radiation, fast precipitation responses were divided into changes caused by SW and LW radiative forcings individually. The basis of SAI is that aerosol fields in the stratosphere reflect solar radiation back to space. Therefore less SW radiation is being absorbed by the background atmosphere below the aerosol layer, leading to an increase in global mean precipitation. However, aerosols themselves also absorb LW radiation, which decreases global mean precipitation. Therefore, in the case of SAI, the fast precipitation change is a tug-of-war between these two components. From this analysis, we can understand the systematic patterns mentioned above: 1) The fast precipitation response was consistently more negative in simulations using M7 compared to SALSA, because M7 producing fewer and larger particles than SALSA, resulting in lower SW radiative forcing (allowing more SW radiation to reach the atmosphere for absorption) but higher longwave (LW) radiative forcing (resulting in more radiation being absorbed) compared to SALSA at the corresponding injection rate (Fig. 4). 2) As demonstrated by Laakso et al. (2022), particles become relatively larger with larger injection rates. Larger particles absorb more LW radiation, but there are relatively fewer smaller and less efficient scattering aerosols for SW radiation. Therefore, LW radiative forcing was rather linear with the injection rate, whereas SW radiative forcing saturated with larger injections. From a precipitation perspective, this means that the LW component (precipitation decrease) becomes relatively stronger against the SW component (precipitation increase) with larger injections.

Relatively minor differences in the radiative forcing of SAI in Fig. 2, in spite of the implementation of identical optical properties, and small differences in the absorption within background atmospheres across ESMs play a significantly larger role in the differing fast precipitation responses between models than one might initially expect. The decrease in absorbed SW radiation, due to the scattering from the stratospheric aerosol field, is fairly close, but with an opposing sign to the absorbed LW radiation (see supplement Fig. S14-15 and Table S1). As a result, the changes in total absorption were less than $\pm 1$ Wm$^{-2}$ for nearly all simulations, with the exception of those involving the most extreme injection rates with M7 aerosols (supplements a and b). In this context, even a slight variation in absorption changes due to SAI across ESMs can have a relatively large impact. For example, while the variation in LW ESMs was minimal, the reduction in SW absorption was 0.12-1.57 W/m$^2$ smalle in simulations using MPI-ESM compared to those conducted with CESM and EC-Earth. Consequently, the total absorption in MPI-ESM simulations was greater than in the other two ESMs, particularly at higher injection rates. This led to a more pronounced negative fast precipitation response in MPI-ESM relative to the other two models. The reasons for differences in absorption of radiation, or radiative forcing, among ESMs are not entirely clear. However, they may be influenced by properties

of the background atmosphere and surface, such as clouds, albedo, aerosols, and gaseous components. Model features and simulation characteristics, like resolution, interpolation of SAI fields, differences in radiation schemes, or how these schemes are integrated with the atmospheric model, might also play a role. An in-depth analysis of these factors is beyond the scope of this study. In equilibrium simulations (see Fig. 6), variations in precipitation responses across the ESMs are influenced also by disparities in the fast precipitation response to $CO_2$ and the radiative forcing of $CO_2$ (Fig. 5). In these simulations, the radiative forcing of $CO_2$ also determines the SAI injection rate, which varies for each model (Fig. 6a)

The findings presented in this study, as well as Part 1, illustrate that variations arising from the microphysical scale and the modeling of microphysical processes can result in substantial discrepancies in the global-scale climate impacts of SAI. This highlights the significant uncertainty that microphysics introduces into our estimations of SAI impacts. Therefore, greater effort should be made to improve the representation of microphysical processes in stratospheric conditions and to understand the observed differences in results between aerosol climate models (Quaglia et al., 2023).

The analysis presented here was largely based on the quasi-linear assumption of a relationship between near-surface temperature and radiation or global mean precipitation change in the case of an abrupt change in the forcing agent. As it is generally known and demonstrated here, this assumption does not completely hold, especially for simulations spanning decades. Even though the method is not perfect, the analysis was consistent across all models used here and proved to be a useful tool in analyzing the factors behind simulated responses.

This study only covers continuous equatorial injection within the longitude bands examined in Laakso et al. (2022) (referred to as the baseline scenario). In Laakso et al. (2022), we simulated various alternative injection strategies involving different magnitudes, and temporal and spatial injection patterns. Many of these alternative scenarios were found to be more effective strategies to scatter SW radiation and absorb less LW radiation than the baseline scenario used in this study. For instance, in the seasonal injection scenario examined in Laakso et al. (2022), which involved seasonal changes to the injection area, the simulated SW radiative forcing at an injection rate of 20 Tg(S)yr$^{-1}$ with M7 was 30% greater than in the injection scenario examined here. However, the difference in LW radiative forcing was small between the two injection scenarios. If similar climate equilibrium simulations, as we did here with baseline injection strategy, were done with seasonal injection strategy, a smaller injection rate would be required. Simultaneously less LW radiation would be absorbed and thus it would result smaller reduction in global mean precipitation than we saw in Fig. 6. Seasonal injection strategy also would probably lead to a more equal compensation of temperature change across latitudes and lesser warming in the Arctic region in climate equilibrium-style simulations since the forcing would be more concentrated in mid-latitudes than the tropics compared to equatorial injections. Simulating different injection strategies with ESMs is a subject for future research.

The overall results of this study indicate that there are significant uncertainties regarding the estimated impacts of the possible deployment of SAI (e.g the coefficient of variation of the fast precipitation response below injection rate 50 Tg(S)yr$^{-1}$ was above 1.5). There are large discrepancies in global mean responses of radiative forcings, temperature, and precipitation, as well as the required amount of sulfur to achieve a certain target, depending on the aerosol and Earth System Model used. These quantities are essential for any consideration related to solar radiation management, and the large uncertainties regarding them raise concerns about the more uncertain quantities, such as regional responses or extreme climate impacts under SAI.

These findings underscore the urgent need for further research on SAI and the development of better tools to analyze and understand the possible impacts of SAI. In its current state, our understanding of the potential consequences of SAI is insufficient to seriously consider implementing these techniques in the near future.

*Data availability.* The model data for this paper are available online (https://doi.org/10.57707/fmi-b2share.36dce66f9e4d44d0972a411a5ab0938b)

*Author contributions.* AL designed the research, performed the experiments, carried out the analysis, and prepared the paper. All authors contributed ideas, participated in interpretation and discussion of the results, and contributed to writing the paper.

*Competing interests.* The authors declare that they have no conflict of interest.

*Acknowledgements.* This research has been supported by the Tiina and Antti Herlin Foundation (grant no. 20200003).

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
