# Peer review of "Dependency of the impacts of geoengineering on the stratospheric sulfur injection strategy – Part 2: How changes in the hydrological cycle depend on injection rates and model?"

_EGUsphere, 2023_

## Author Comment (AC1)

We are grateful to Peter Irvine for his comprehensive review, insightful suggestions, and valuable comments, which have enhanced the quality of our manuscript and made it clearer and easier to understand. Our point by point answers to the comments are presented below. Referee comments are in bold and our replies in body text.

**General comments**

**The authors evaluate the precipitation response to stratospheric aerosol injection (SAI) geoengineering, considering earth system model and aerosol microphysical uncertainty. Prescribed aerosol fields were generated in an ESM with either a sectional or modal aerosol module, producing quite different aerosol properties and hence radiative forcings. These were fed into 3 different ESMs which simulated a range of combinations of CO2 and SAI injections. The fast, forcing-driven hydrological response was found to be quite different for the different aerosol modules as the modal module produced fewer, larger particles which absorbed more LW radiation. Despite being driven by the same aerosol field, the ESMs produced quite different radiative, temperature and precipitation responses. However, the largest differences in many respects arose from the microphysical representation. The study makes a detailed analysis of the various factors that shape the precipitation response to SAI, making clear that microphysical uncertainties are important.**

**This paper will make a substantial contribution to the literature, is generally well-written and has generally good quality analysis, and so I recommend that it be published after making relatively modest changes, outlined below.**

**The paper is generally clear and well-written, but the argument was a little hard to follow in places as the paper jumped back and forth between radiative forcing and precipitation several times. For example, section 4.4 is titled "simulated precipitation response…" but the opening page is about the reasons for a radiative mismatch. The authors may consider revising the order of analysis.**

To make results section more clear, results section is divided into two subsection:

*"4.1 Quantifying fast and slow responses from regression simulations"*

*"4.2 Results of climate equilibrium simulations"*

First two paragraphs in prior section 4.4, which were describing climate equilibrium simulations and how SAI-CO2 pairs are chosen, are moved under new section 4.2.

Section 4.4 is now divided to two subsubsections:

*"4.2.1 Global mean temperature change in climate equilibrium simulations"*

and

*"4.2.2 Simulated global mean precipitation change in climate equilibrium simulations"*

**The figures and analysis are generally very good, but in places the analytical choices made things a little difficult to follow, e.g., Figure 6 was particularly challenging. I've made a series of suggestions for improvement in the specific comments below.**

**I was left not quite knowing the answer to a question that I think could help increase the impact of this study and I think that with a little work could be easily answered. There is a factor of ~2 difference in the SO2 amount needed to achieve the same cooling for the sectional and modal aerosol modules. This made me wonder: is the residual precipitation, or just fast precipitation, difference ~2x larger as well? Or does the fast effect of CO2 dominate this residual?**

We have included two new figures in the supplement to answer these questions. In these figures we focus on the uncertainties caused by SAI and exclude the impact of CO2 in these figures, as the main focus of this study is on the impact of SAI.

Fig.6 results were for simulations where magnitude of SAI is adjusted to compensate for CO2 induced warming. There, CO2 fast response dominates the global mean precipitation, but the SAI precipitation component is significant in most cases. To show this, the following figure is added to the supplement, which shows the fast precipitation component of SAI in scenarios shown in Fig. 6b. So for example, the fast precipitation component of SAI varies from -0.9 to +0.7 % depending on the aerosol model and the ESM in cases where the warming is caused by 600 ppm CO2. The CO2 fast precipitation component for 600 ppm CO2 is -3.0 - 1.9  % depending on the ESM (see Fig 5). In addition we have included a figure showing the  fast precipitation response of SAI with radiative forcing as x-axes (left Figure below). As this figure shows, the connection between fast precipitation response and radiative forcing is complex in individual models, but generally uncertainty (difference in results between models) increases with larger injections. This figure also justifies some of our analytical choices, which will be discussed in more detail later in the specific comment.

[Figure]

Related to the new supplement figure, the following text is added to manuscript in the first paragraph of section 4.2.2: *"...and the differences between aerosol models become more pronounced with higher injection rates. These differences among aerosol model results are even more apparent when the fast precipitation response is presented as a function of radiative forcing. For SALSA aerosols, a lower injection rate can achieve the same level of radiative forcing as M7, resulting in more significant differences in fast precipitation responses (see supplement figure Fig. S7a)."*

And in section 4.1.3 it now reads: "*As illustrated by Fig. 5b and Supplementary Fig. S6, the fast precipitation response to a quadrupling of CO2 levels varied significantly, ranging from a decrease of 3.38% in the EC-Earth simulations to an increase of 5.6% in the CESM simulations. However, the fast precipitation response to SAI accounts for differences of up to 3.5% in global mean precipitation, as illustrated by the fast precipitation response component in Supplementary Figure S7b.*"

Another new figure is related to the following comment and shows how uncertainties depend on injection rate or radiative forcing:

**More generally, could the authors comments on the relative scale of the precipitation differences compared to this injection amount? RMSE difference might be a simple metric that could be calculated to test this. Some take-away claim that relates these 2 key elements would make the paper more memorable and useful to the community.**

New figure (S8) on standard deviation is added to supplement:

[Figure]

*Figure S8. Standard deviation (SD) of simulated a) radiative forcing at TOA and b) fast precipitation response as a function of injection rate, c) the coefficient of variations of radiative forcings and d) fast precipitation responses and SD of fast precipitation responses as a function of injection rate. Since specific radiative forcings marked in d) were not simulated, the fast precipitation responses are estimated based on the two closest simulated radiative forcings for each model.*

Following text is added to the manuscript to section 4.1.2:

*However, the standard deviation of the simulated fast precipitation response between model combinations is rather linear with respect to the injection rate and the simulated radiative forcing (see Supplement Fig. S8). This means that the differences in the simulated fast precipitation response between models become larger with larger injections.*

**Specific comments**

**L14 – reduction relative to what?**

Line now reads: "*In equilibrium simulations, where aerosol injections were utilized to offset the radiative forcing caused by an atmospheric $CO_2$ concentration of 500 ppm, the decrease in global mean precipitation varied among models, ranging from -0.7% to -2.4% compared to the preindustrial climate.*"

**L16 – "rather negatively correlated" – why not just negatively correlated? And could you clarify what is meant by "absorbed radiation" here? Is that a new finding or a widely established result that you are referencing?**

Word *"rather"* is replaced by "i*s shown to be* ". The relation between fast precipitation response and atmospheric absorption is not a new finding. It is already shown e.g in here: Samset, B. H., G. Myhre, P. M. Forster, Ø. Hodnebrog, T. Andrews, G. Faluvegi, D. Fläschner, M. Kasoar, V. Kharin, A. Kirkevåg, et al. (2016), Fast and slow precipitation responses to individual climate forcers: A PDRMIP multimodel study, *Geophys. Res. Lett.*, 43, 2782–2791, doi:[10.1002/2016GL068064](10.1002/2016GL068064)

Good point related to *"absorbed radiation"*, because this claim in the manuscript was actually wrong. It is not correlated with absorbed radiation, but absorbed radiation by the atmosphere. "absorbed radiation" is replaced by "*atmospheric absorption*".

**L30 – relative to what?**

This is now rewritten as: "*Thus, this would lead to several consequences such as a decrease in global mean precipitation and unevenly distributed temperature chance compared to climate without increase in CO2 and SAI*"

**L31-34 – review phrasing.**

These lines and the next sentence are now rewritten as: "*The extent of these impacts is influenced not only by the level of GHG increase in the atmosphere but also by the interaction of aerosol fields with SW and LW radiation. This interaction is further dependent on the aerosols' optical properties, which, as demonstrated in Laakso et al. (2022) study, are closely associated with the modeling of aerosol microphysics in climate models.*"

**L43 – clarify whether the same 2 aerosol modules were used in the 3 different models.**

*This now reads: "In Laakso et al. (2022) (from now on referred as Part 1), we simulated different injection rates using both a sectional aerosol model SALSA and a modal model M7 within same climate model ECHAM-HAMMOZ, showing that…"*

**L50 – might be nice to indicate roughly the fractional changes here.**

*This now reads: "This means that in larger injection rates the contribution of LW radiation to total radiative forcing becomes larger: In SALSA simulations LW radiation forcing compensated for between 10% to 28% of the SW radiative forcing with injection rate of 1-100 Tg(S)yr-1 while M7 simulation the range was 24-57%."*

**L23-50 – Might be worth indicating which aerosol scheme performs better at reproducing observed volcanic response if that can be determined, i.e., is the SALSA sectional model better but more expensive and M7 the poor-man's alternative?**

Following discussion is added to text:

*The situation is further complicated by the lack of clear criteria for selecting the appropriate aerosol model. Observations following the 1991 Pinatubo eruption have frequently been utilized as a benchmark for evaluating models' ability to simulate stratospheric aerosols. However, a single sulfur injection, as in the case of Pinatubo, differs significantly from continuous injections in case of SAI. Notably, there is a minimal difference between the M7 and SALSA model results in the simulations of the Pinatubo eruption, as detailed in Kokkola et al. (2018). Simulations using the M7 model were 60% faster than those with SALSA, but, there were some numerical limitations associated with the modes in M7, which restricted the aerosols from achieving an optimal size range for effectively scattering radiation, as noted in Laakso et al., (2022). However, the performance of the M7 results is also sensitive to the configuration of the modes, making it difficult to predict which setup will perform well, as the performance depends on the simulated case (i.e volcanic eruption vs. SAI, different injection strategies for SAI).*

**L59-60 – Does this apply in the same way to stratospheric heating as it does to tropospheric? Is stratospheric heating as effective as tropospheric heating at suppressing precipitation? If the absorption occurred up in the mesosphere, I imagine it would have little effect on the hydrological cycle.**

Based on Fig 4a, it does apply (e.g. $CO_2$ (affects both the troposphere and the stratosphere) and SAI (stratospheric) impact are similar - the dependence between absorption and precipitation is the same). This is how we see that it can be understood:

The atmosphere has a relatively low heat capacity, meaning it quickly adjusts to changes in energy fluxes. This adjustment ensures a balance where incoming and outgoing energy fluxes in and out from the atmosphere are equal. When some factor increases the absorption of energy in the atmosphere (i.e., the energy flux into the atmosphere), it rapidly adapts to this and will have a new equilibrium. This adaptation involves an increase in temperature, which leads to more longwave (LW) radiation being emitted, and/or a decrease in latent heat flux, seen as reduced precipitation.

Atmosphere can be separated into two layers: the stratosphere and the troposphere which both have to be in equilibrium in respect to energy in a relatively short time scale. In these simulations absorption of the stratosphere is increased due to aerosols. Since the stratosphere primarily exchanges energy with adjacent layers (and thus the troposphere) through radiative fluxes, it must warm up to emit more radiation and achieve a new equilibrium. This warming results in more radiation being emitted towards the troposphere, thereby increasing the incoming LW radiative flux to the troposphere. Consequently, the

troposphere adjusts to this new influx by decreasing its latent heat flux, finding a new balance in the process.

Niemeier et al. (2013) investigated the impact of different SRM techniques acting at different altitudes. They made a similar comparison as we have done here, comparing predicted precipitation (mainly based on absorption) and simulated precipitation. As these results were in good agreement, this suggests that it does not matter at what height the absorption occurs.

In the introduction there now reads: *"Niemeier et al. (2013) investigated the impact of different SRM techniques applied at different altitudes. Their findings show that the precipitation predicted by Equation 1 aligns closely with the precipitation observed in simulations. Changes in sensible heat flux within their simulations were minimal, suggesting that the calculation of precipitation based on atmospheric absorption is not influenced by the altitude at which the absorption change occurs."*

Please also note replies for another reviewer.

**L64-66 – perhaps note T-driven intensification under GHG case?**

These lines now read:

*"In the case of solar radiation modification generally, the unambiguous impact of this is seen in model simulations, in cases where the GHG-induced radiative imbalance is fully compensated by SRM. Without SRM, there would be an increase in global mean precipitation, driven by a rise in temperature. However, if the temperature increase is offset by SRM, it results in overcompensation and decrease in global mean precipitation"*

**L75 – formatting of citations.**

Fixed

**L78 – in the consequent precipitation responses.**

"followed" changed to "consequent"

**L103 – add resolution in degrees.**

Resolution in degrees is added to line: *" The resolution of atmosphere used in MPI-ESM, CESM and EC-Earth simulations are T63L47 (1.9◦ x 1.9◦), finite volume 0.9◦x1.25◦ and 32 vertical levels and T255L91 (0.70◦ x 0.70◦ ) respectively"*

**L149 – from a preindustrial baseline with GHG and SAI perturbations applied?**

Line changed to: *"The regression simulations with ESMs were started from a preindustrial baseline with GHG and SAI perturbations applied"*

**Figure 1 – Great figure! Small suggestion: 6x climate responses instead of impacts.**

Thank you! *"impacts"* changed to *"responses"*

**L162 – logarithmic fit**

*"logarithmical"* changed to *"logarithmic"*

**Figure 2 – Another great figure. Wondered if it might make sense to use the shape to match models, e.g., diamonds = CESM. This might help the colorblind to follow along. Looks like that was done in Figure 3, but I'd suggest adding the shapes to the legend or caption.**

Shapes are now matching with ESMs, simulations where SALSA aerosols are used are now surrounded by black edges and legend is changed in Fig 2,3,4,6 (shapes are changed also in Fig3). Example of the new "style" is seen in the first figure above which were added to the supplement. This was a good suggestion and now it is much easier to distinguish different scenarios.

**L206 – will have changed when it does settle down?**

This line now reads: *"...it is possible to estimate how much global mean temperature will have changed when it does settle down in the new radiative balance after SAI is started…"*

**L205 – 213 – a little repetitive.**

We feel that this part of the text still provides information which has not been mentioned elsewhere in the manuscript.

**L245-249 - phrasing a little unclear.**

These lines are now rewritten as: *"Thus, when the effective climate sensitivity is calculated based on 150-year simulations, the sensitivity appears higher in the CESM model compared to the other two ESMs. However, this difference is not as pronounced when using 20-year simulations. This characteristic in the CESM results was discussed in Bjordal et al., 2020. It was identified that the increased sensitivity is due to a negative feedback mechanism, which involves a reduction in ice content within clouds in a warming climate. This feedback mechanism becomes less substantial when the climate has warmed sufficiently."*

**Figure 4 – Is it best to compare injection mass for Salsa and M7 directly in this way? I found myself a little confused until I remembered that 50Tg in Salsa has a much greater cooling effect than in M7. Perhaps some additional text or analysis could clarify this, e.g., normalizing the fast effect by the expected cooling magnitude or plotting against an x-axis that shows temperature or RF?**

We think that this is dependent on the perspective of the reader. You are correct in saying that for certain types of "end-users," it might be more useful to understand the extent of precipitation changes associated with a specific degree of cooling (radiative forcing), which is often the premise for consideration of SAI. This was the perspective we chose to follow e.g in figure 6 where CO2 concentration is chosen as x-axis. However, in this instance, we adopted a more of a climate modelers' perspective, where the same perturbation (specific injection rate) is applied to different models to compare the variations in their responses. Using injection rate as x-axes is also inline in earlier figures related to temperature and forcing as well as figures in Part 1.

Figure 4 is just for the explanation responses seen in Fig3b. Thus we see that x-axes should be the same here. However, as said in our earlier comment, we added a new figure in the supplement, where fast precipitation response is shown as a function of radiative forcing and this was commented on in the text.

**L295-296 – Would this non-linearity disappear if the axis was RF instead?**

Please see our earlier comment.

**L347 – less precipitation = a greater reduction in precipitation relative to the baseline?**

Changed as suggested

**L350 – link back to earlier claim on reduced SO2 for same RF in EC-earth?**

We added *"This is due to two factors in EC-Earth simulations: the smaller magnitude of the fast precipitation response to CO2 (as shown in Fig. 5b) compared to MPI-ESM and CESM, and the more positive fast precipitation response to SAI when the injection rate is adjusted to match the radiative forcing of CO2 (refer to Fig. S7b)."* to text.

**Figure 6 – Is this the best way to get this information across? I'm very confused by some of the analytical choices and by how complex it is. Why aren't the points falling on the precise CO2 ppm values used before? Can the analysis be flipped so that they do?**

If these kind of simulations are done for certain CO2 ppm and SAI pairs, where forcings of these two compensate each other, there are two choices how to proceed:

a) Choose specific CO2 concentrations and adjust injection rate to correspond forcing of CO2

or b) Choose specific injection rates and adjust CO2 concentration to correspond forcing of injection rate

We have chosen to proceed with the second option. As Figure 4 demonstrates, both the radiative forcing and the fast precipitation response exhibit a logarithmic dependence on CO2 concentration. Therefore, it is straightforward to calculate values between simulated CO2 levels. However, adjusting SAI levels is not as simple, particularly for the fast precipitation response. In this case, fitting any simple function to the simulated points is challenging, as can be seen in Figure 2b or the new Figure S7 in the supplementary material.

This was commented earlier on in the first paragraph in section 4.4. The following text is also added now to caption of Fig6: *"Based on the logarithmic relationship between radiative forcing and fast precipitation response to CO2 concentration (as shown in Fig. 5), the CO2 concentration and the subsequent fast precipitation response can be determined from the logarithmic fit so that the radiative forcing aligns with the simulated radiative forcing for SAI."*

(continuation related to fig6) **More information needed on c, to clarify modelled pairs. Panel d seems like it could have been a whole multi-panel figure of its own. I also**

**wonder if a pure temperature adjustment is the best choice, couldn't you also scale up or down the fast effect of SAI by the fractional change in cooling that's needed? Presumably that would give a better fit.**

In addition to the new line in above to caption fig6 we modified the caption: *"c) Simulated changes in a) global mean temperature and b) precipitation under SAI - CO2 pair scenarios (as illustrated in a), assuming a state of climate equilibrium."*

We acknowledge that d) includes a lot of information and takes some effort to comprehend. However, to compare fast responses, modeled precipitation, and temperature adjustments easily they need to be in the same panel. An alternative approach would have been to allocate separate panels for each Earth ESM, but our aim was to facilitate direct comparisons of actual simulated results between models.

You are correct about adjustments. A pure temperature adjustment would not be enough if we would like to estimate what would be "real" precipitation change in the case without temperature change at all. However, here we want to show that the assumption on total precipitation based on fast responses in Fig6 b) did not correspond to the modelled values mainly because of an unexpected temperature change. I.e precipitation change should be $\Delta P = a*\Delta T + P_{fast-CO2} + P_{fast-SA}$ste, where we did not take into account "slow response" ($a*\Delta T$). By removing "slow response" from the simulated precipitation values, there should be only fast responses of SAI and CO2 left and it should correspond to the calculated precipitation values in Fig 6b. Even though there was temperature change in actual simulation, the fast effect of SAI is still the same. Scaling might become a point of discussion if we aimed to estimate the precipitation levels in a hypothetical equilibrium state without temperature changes. However, that was not our intention here. In addition, as previously mentioned, scaling the fast precipitation response is not straightforward due to the nonlinear relationship between fast precipitation response and radiative forcing.

 *"Adjusted by hydrological sensitivity"* might have been a bit misleading and now *"Adjusted by hydrological sensitivity"* in the legend to *"slow (temperature) response removed"*.

**L355 – conversely? should that be Additionally?**

Yes it should. Fixed

**355-360 – this suggests switching axes on Figure 6, as CO2 is the dependent variable.**

You're correct. However, our intention was to adopt a slightly different perspective and illustrate the significant uncertainties involved in mitigating CO2-induced climate change through SAI. We aim to highlight how these uncertainties, particularly the differences in simulated values, escalate with the magnitude of climate change.

**4.4 – Given the first page is about the radiative mismatch, should this be 2 sub-sections? And should the radiative discussion come here or earlier? This might help with the flow of the article.**

This section is now separated to two sections: "*Global mean temperature change in climate equilibrium simulations*" and "*Simulated global mean precipitation change in climate equilibrium simulations*"

**L392 – global mean precipitation is more positive?**

*"larger than"* changed *"more positive than"*

**L393 – here you are referring to the effect after the fast effect, whereas in some studies it is meant to include the total effect.**

"*…than the estimated ones (which did not take into account precipitation change due to the hydrological sensitivity and change in the temperature)*" is changed to: *"..than those estimated from the sum of fast precipitation responses"*

**L398-400 – I think making the correction I suggested and noting that the forcing mismatch produced this precipitation mismatch might lead to a more useful conclusion here.**

It was a good and justified suggestion. However we decided to keep the figure and these lines as it is. As replied to earlier comment, here we want to compare $\Delta P = a*\Delta T + P_{fast-CO2} + P_{fast-SAI}$ to actual simulated precipitation. Even though our assumption that there would not be temperature change was wrong for CESM and EC-EARTH, the fast precipitation components should still be the same.

**L403-420 – Isn't a big driver of the overcooling / residual warming seen in many stratospheric aerosol geoengineering experiments the distribution of aerosols? Might be useful to refer to that distribution here and remind the reader that it's the same in each model (I may have forgotten myself by this point).**

Yes, the distribution of aerosols significantly contributes, but overcooling or residual warming is also observed in experiments where the solar constant is reduced (e.g Schmidt, et al., (2013)). This outcome is somewhat anticipated when there's a fractional reduction in solar radiation coupled with an increase in well-mixed CO2 and because the average gradient between high and low latitudes is steeper for solar radiation than for thermal radiation. Although the overcooling/residual warming in the case of solar constant reduction was much smaller than the SAI based on e.g Visioni et al. (2021).

We include the following text to the manuscript:

"*Laakso et al. (2022) demonstrated that the radiative forcing from SAI is primarily concentrated around the Equator for aerosols simulated using both SALSA and M7 models. There was also significant clear-sky zonal forcing observed at the latitudes of 50◦N and 50◦S. However, the presence of clouds in these regions reduced the aerosol all-sky radiative forcing. Aerosol optical properties were consistently applied across all three ESMs, but variations in cloud cover and properties among the ESMs can lead to differences in the actual radiative impact of aerosols.*"

Schmidt, H., Alterskjær, K., Bou Karam, D., Boucher, O., Jones, A., Kristjánsson, J. E., Niemeier, U., Schulz, M., Aaheim, A., Benduhn, F., Lawrence, M., and Timmreck, C.: Solar

irradiance reduction to counteract radiative forcing from a quadrupling of $CO_2$: climate responses simulated by four earth system models, Earth Syst. Dynam., 3, 63–78, https://doi.org/10.5194/esd-3-63-2012, 2012.

Visioni, D., MacMartin, D. G., Kravitz, B., Boucher, O., Jones, A., Lurton, T., Martine, M., Mills, M. J., Nabat, P., Niemeier, U., Séférian, R., and Tilmes, S.: Identifying the sources of uncertainty in climate model simulations of solar radiation modification with the G6sulfur and G6solar Geoengineering Model Intercomparison Project (GeoMIP) simulations, Atmos. Chem. Phys., 21, 10039–10063, https://doi.org/10.5194/acp-21-10039-2021, 2021.

**Figure 7 – maybe a note on how these pairings were chosen. It might be useful to extend the y axis and add a global mean temperature residual value to the legend.**

Text *"In these simulations, the CO2 concentration was adjusted to counterbalance the radiative forcing from a specific injection rate, as determined by regression simulations."* was included in the caption of Figure 7.

Figure 7 was modified as suggested and in the caption it now reads: *"dT in the legends shows residual global mean temperature"*

**Figure 8 – A bit difficult to read, would adding figure wide column and row labels make it easier to parse? You might also consider rearranging so that SALSA is as one block, M7 as another.**

SALSA and M7 are moved to their own blocks and the aerosol model label is moved to the side of the figure. ESM is now the label for the whole row. The title of each panel includes only the injection rate and CO2 concentration and the residual global mean temperature. "-piControl" is removed from the panels.

**Figure 9 – missing labels. Panel a is quite difficult to read, some for previous figure. Is there another way to show this?**

We acknowledge that panel a is somewhat challenging to read. However, we have not come up with an idea of a better method of presentation, so the figure has been retained in its current form.

**Figure 10, same comment as 8.**

Modified as figure 8.

**489-494 – not particularly clear or particularly logical flow at the end of this paragraph, consider revising.**

We removed the line *"Thus it is important to bear in mind when interpreting these results, but also in general, not to assign excessive importance to the quantified effective climate sensitivity of individual models, as it is sensitive to external factors (e.g., forcing agent and simulation period)."*

from 489-> and end of this paragraph it now reads:*"Overall, drawing from these results and a comparison with the climate sensitivities reported in Zelinka et al (2020), it should be kept in*

*mind that effective climate sensitivity is not a straightforward parameter. Its interpretation is complicated by its sensitivity to external factors, such as the type of forcing agent (which affects shortwave vs. longwave radiation) and the length of the simulation period (e.g., 20 years vs. 100 years). Moreover, sensitivity to these external factors varies across different models."*

**504-505 – compared to what? Is the comparison to the baseline the most relevant? Should it be to the 500 ppm case? Given the amount of SO2 injected scales with CO2, this difference in injection amount should modulate that total precipitation response, which as a consequence shifts the net result.**

Now it reads: *"In the aforementioned 500ppm atmospheric CO2 concentration and SAI scenario the resulting reduction compared to preindustrial climate in global mean precipitation ranged from 0.7% to 2.\% between different model combinations."*

We agree that it probably would be more convenient to compare a situation without SAI (500 ppm) than a pre industrial climate. However to simulate precipitation change in 500 ppm would need a very long simulation where climate was near new climate equilibrium.

**L513-514 – See my earlier comment about making a full adjustment, i.e., what would have occurred if the correct amount had been chosen to keep temperature constant, rather than just the temperature adjustment (which excludes the change in fast forcing effect).**

In addition to our previous response regarding the complexity of scaling the fast precipitation response to correct for the necessary radiative forcing to eliminate residual temperature change, estimating the required radiative forcing is not straightforward. As demonstrated by these simulations, it was not feasible to counteract warming by adjusting CO2 radiative forcing to offset the effects of SAI. Furthermore, the Gregory plots provided in the supplementary materials for these 'climate equilibrium simulations' do not show a distinct 'residual radiative forcing'. Therefore, making a comprehensive adjustment would likely necessitate the use of some form of feedback function in the simulations to correct the CO2 levels (or SAI).

**L495-513 – Here or elsewhere some comment on the relative scale of the precipitation differences compared to the required injection amounts would be useful. M7 suggests ~2x greater sulphate required, is the gross or net precipitation difference 2x greater too?**

There is now a new figure in the supplement (Fig. S8, see our earlier reply) showing the standard deviation of the fast precipitation response as a function of radiative forcing. It does indeed show that the differences in fast precipitation responses increase relatively linearly as a function of injection rate. Comment on this is added to section 4.1.2 (see our earlier reply).

**518 – more negative?**

*"lower"* changed to *"more negative"*

**530 – consistently more negative?**

*"lower"* changed to *"more negative"*

**538 – perhaps remind reader that they faced the same change in aerosol optical properties**

Now it reads: *"Relatively minor differences in the radiative forcing of SAI in Fig. 2, in spite of the implementation of identical optical properties, and small differences.."*

**543-547 – a little hard to follow.**

Now it reads: *"For example, while the variation in LW ESMs was minimal, the reduction in SW absorption was 0.12-1.57 W/m2 smalle in simulations using MPI-ESM compared to those conducted with CESM and EC-Earth. Consequently, the total absorption in MPI-ESM simulations was greater than in the other two ESMs, particularly at higher injection rates. This led to a more pronounced negative fast precipitation response in MPI-ESM relative to the other two models"*

---

## Author Comment (AC2)

We thank the reviewer for suggestions and comments. Comments helped to clarify several parts of the text. Our point by point answers to the comments are presented below. Referee comments are in bold and our replies in body text.

**Reviewer 2:**

**The paper provides an extensive analysis of inter-model difference in the global (and regional) precipitation response to Stratospheric Aerosol Injection. On the whole, this is a sound piece of work, carefully analysed and well written, with clear plots.**

**. On the whole, this is a sound piece of work, carefully analysed and well written, with clear plots.**

**A not too major criticism is that the paper is rather technical, and readability and possibly usability by a larger set of readers could be improved by adding some clarifications and physical interpretation here and there. I hope my comments can help. In addition, there is a handful of minor improvement points regarding things like figure captions and labels, listed below.**

**Interpretation and readability**

**Overall aims: Maybe the overall aim(s) could be stated in a small number of clear research questions at the end of section 1. Currently line 81 states: "We investigate how these impacts depend on the injection rate and the aerosol microphysics model", which is relatively vague and mixes physics questions (how does precipitation change under different SAI intensities) with modelling questions (is there model uncertainty). Unless the main focus is strictly the model uncertainty part, the paper, which is now relatively technical, may profit here and there from a bit more physical interpretation.**

Now it reads: *"We investigate how aerosol impact on SW and LW radiation changes the atmospheric absorption and further atmospheric energy budget and hydrological cycle. We also study how precipitation changes under different SAI intensities. Furthermore, we examine how these outcomes vary based on the aerosol microphysics model employed to simulate the aerosol fields, as well as the Earth System Model used to simulate climate responses."*

**Structure, especially Section 4: It would help the reader to get a short hint at the beginning of sect 4 what the subsequent pieces of analysis are meant to do and how they relate to each other and the overall aim / research questions of the paper. For example, it helps to know before sect. 4.3 that you first estimate the precip change based on the radiation diagnostics and then will compare the overall change to the actual model results in 4.4.**

*This was a good suggestion. Now in the beginning of results section it reads:*

*"In this section, we begin by employing regression analyses on simulations to estimate the temperature changes in simulated SAI scenarios, based on the effective climate sensitivity. We then proceed to quantify the fast precipitation response and the radiative forcings*

*associated with simulated SAI and CO2 perturbations. These metrics allow us to estimate the extent of CO2 radiative forcing that each simulated SAI scenario could offset. Given the assumption that there should be no change in global mean temperature, the quantified fast precipitation responses can then be utilized to estimate changes in global mean precipitation in scenarios where the radiative forcings of SAI and CO2 are balanced. Lastly, we conduct climate equilibrium simulations for various SAI injection rates and their corresponding CO2 concentrations. These simulations are utilized to examine how estimated precipitation changes, based on the fast precipitation responses, differ from the actual simulated values and to analyze regional responses."*

**Fast response and absorbed radiation, line 56 ff. The paragraph could be clearer. Line 57 "Further precip change" -> further with respect to what? Line 59: rather than saying "Any change in X translates to a change in Y", it is clearer to say e.g. "Any increase in X translates to a decrease in Y", to immediately give the direction of change. The whole sentence seems unnecessarily long-winded. Most importantly, since some of the readers may not be experts in hydrological cycle but e.g. in SRM or impact modelling, it would be helpful to explain in a bit more detail 1) what fast and slow precip responses are (you mention the fast one but not what the slow one is) and 2), give a few sentences about the physical meaning of the link between absorption of radiation (up to which height?) and the global precip response. I appreciate you give several references, but seeing how central this information is to the whole paper, it increases readability of the piece to spend a few more sentences (and an equation or two).**

We rewrote the paragraph about atmospheric energy budget and fast and slow responses in the introduction section so that now it reads:

*"Changes in atmospheric radiation have a direct impact on precipitation. Precipitation changes can be explained by the changes in the total column atmospheric energy budget (O'Gorman et al., 2012). The atmosphere possesses a relatively low heat capacity, and following a perturbation, it rapidly reaches a state where the incoming and outgoing energy fluxes to and from the atmosphere balance each other. In other words, the budget of perturbations between two atmospheric states can be expressed as:*

$$L\delta P = \delta RSurf - \delta RTOA + \delta SH = -\delta Rabs + \delta SH, \quad (1)$$

*where $L$ is the latent heat of condensation, $P$ is precipitation, $RTOA$ and $RSurf$ are the change in the radiative fluxes at the top of the atmosphere and surface, $SH$ is the sensible heat flux change and $\delta Rabs$ is the change in absorbed radiation. Niemeier et al, 2013, showed that changes in global latent heat flux dominate changes in sensible heat flux, establishing a roughly linear relationship between precipitation and the discrepancy between the radiative imbalance at the surface and at the top of the atmosphere. Other studies have also shown that changes in precipitation are proportional to the difference between changes in radiation at the surface and in the atmosphere, i.e absorbed radiation (O'Gorman et al., 2012; Kravitz et al., 2013b; Liepert and Previdi, 2009). The atmospheric energy budget can also be utilized to represent precipitation in a transient climate. Given that radiation (and changes in atmospheric absorption) are known to be relatively linearly correlated with global mean precipitation, as evidenced by climate models (e.g (Zelinka et al., 2020)) and*

*observations (Koll and Cronin, 2018) precipitation change can be approximated by a simple equation comprising temperature dependent and independent components(s):*

*$\delta P = a\delta T + F = Pslow + Pfast$, (2)*

*where $\delta T$ is the global mean temperature change, a is constant and F are the temperature independent components. Within this equation, $a\delta T$ accounts for all feedbacks attributable to temperature change, including the variation in surface sensible heat flux. This is often referred to as the slow precipitation response or component, which changes over a multi-year timescale alongside alterations in sea surface temperature. F is referred to as fast precipitation response (or component) or rapid adjustment. It can be considered to include the direct radiative forcing, or precisely direct change in absorbed radiation. Thus, at the global scale, a change in global mean precipitation has been shown to be linearly dependent on the absorption part of the induced radiative forcing (Laakso et al., 2020; Myhre et al., 2017; Samset et al., 2016); therefore, a stronger absorption of radiation is linked to a decrease in global mean precipitation"*

And late in the introduction it now reads: *"Niemeier et al. (2013) investigated the impact of different SRM techniques applied at different altitudes. Their findings show that the precipitation changes predicted by Equation 1 aligns closely with the precipitation changes observed in simulations. Changes in sensible heat flux within their simulations were minimal, suggesting that the calculation of precipitation based on atmospheric absorption is not influenced by the altitude at which the absorption change occurs."*

**Following up on the fast response and absorbed radiation relationship (see also eq 1 of O'Gorman 2021 which you cite): I am wondering about the direction of causality. Is it really such that changes in absorbed radiation determine precip, and not vice versa? After all, precipitation (and evaporation) changes may be related to changes in water vapour content or clouds, which may feed back on radiation budget. So it would be good to clarify whether the relationship is (largely) a causal one, or whether it should be seen as merely a diagnostic relationship. If the latter is the case, then of course it can still be used for e.g. the analysis in sect. 4.3, but I would then suggest to me more careful with statements such as "precipitation changes as a function of injection rate can be understood based on the absorbed radiation" (line 279), a formulation which to me suggests causality.**

This is a valid point. While the aerosol fields represent the external variable being modified, and their influence on both shortwave and longwave absorption is logical, it's not definitive based on our results that the latent heat flux wouldn't impact the atmospheric temperature, thereby affecting the emission of LW radiation which would be then see as a change in the atmospheric absorption. However, the e.g observed differences in atmospheric absorption and precipitation between solar dimming and stratospheric aerosol simulations suggest a direction of causality (Visioni et al., 2021). We have now clarified this in the introduction (see our earlier response).

Visioni, D., MacMartin, D. G., & Kravitz, B. (2021). Is turning down the sun a good proxy for stratospheric sulfate geoengineering? *Journal of Geophysical Research: Atmospheres*, 126, e2020JD033952. https://doi.org/10.1029/2020JD033952

**sect 4.2 ff: you focus strongly on the fast precip response. Obviously this is an important quantity, especially in scenarios where GMST changes and hence the slow response are eliminated by SAI. However, since other scenarios are conceivable (e.g., keeping GMST change at 1.5 degrees), it would be quite nice to know how the fast response compares with the slow response. This can be inferred from S6, but is not discussed much. Maybe summarise the results in an equation like "P = a C + b S + c T" where P is the precip change, C the radiative forcing from CO2 (GHG), S the forcing from SAI, and T the GMST change, and a,b,c, are the fit parameters that arise from this study, though admittedly, at least b will suffer from nonlinearities (fig. 4).**

This was a valuable suggestion, and the proposed equation could indeed be informative. However, we opted for an even simpler approach. We have now included the following line in the sections discussing the range of fast precipitation responses for the simulated SAI scenarios:

*"Overall quantified fast precipitation response due to the SAI varied between 0.69% increase in global mean precipitation to -3.19% reduction in precipitation depending on injection rate and ESM-aerosol model combination. Based on the average hydrological sensitivity in our simulations (Supplement Fig. S6), which were 2.46 %K−1 (σ=0.28 %K−1) the range between the maximum and minimum fast precipitation responses corresponds to a global mean precipitation change associated with a temperature variation of 1.6 K"*

**Fig 4a: you state in the main text that the slope differs little among models. However, could you comment also on whether the slope is the (approximately) same for SAI and CO2? at least for MPI-ESM and SALSA, this seems not certain to me from the plot.**

We included a new paragraph at the end of the section 4.3:

*"However, as indicated by Supplement Figure S6, employing a simplistic approach using fast and slow responses to estimate precipitation changes may not be straightforward. Supplement Figure S6 reveals variations in the hydrological responses among the three Earth System Models (ESMs), particularly in the variation of the hydrological sensitivity (i.e., the slope in the figure) across various simulated forcing agents. Simulations using CESM and MPI-ESM suggest that the hydrological sensitivity increases with larger injections. But the range of this increase differs significantly from the sensitivity observed in simulations where CO2 concentration was perturbed. Conversely, in EC-EARTH simulations, hydrological sensitivity ranged from 2.39 to 2.48 %K-1 in scenarios with CO2 perturbations, while in SAI scenarios, the total range was 2.79 - 3.22 %K-1. This discrepancy is a crucial factor to consider, especially in cases where the forcing induced by CO2 and SAI does not fully offset each other but might also have an impact when those are expected to compensate each other."*

**line 373: is there any clear physical reason why GMST increases in two models despite radiative balance being closed?**

There are some indicators for possible physical reasons which can be made based on regional responses in the next section (i.e arctic warming and melting sea ice, different responses of stratocumulus clouds on SW and LW radiative forcing). These topics were

discussed in greater depth in the next section. Initially, this was mentioned at the end of the paragraph where line 373 is found, with the statement, *'We will discuss more about this in section 4.6.'* However, to enhance clarity, it has now been revised to read, *'We will discuss possible physical reasons for the residual global mean warming in CESM and EC-Earth simulations in section 4.2.3."*

**line 395, fig 6d: You suggest that in EC-earth, the correction hydrological sensitivity (i.e. effect of residual GMST change) "slightly overadjusts" precip estimates. This seems rather optimistic. In fact, the error hardly shrinks, of even becomes worse, in some scenarios in EC-earth, even if the correction works nicely in CESM. So it seems to me that in EC-earth there is stuff going on that is not easily captured by your method… could you comment?**

This is a valid point. "slightly overadjust" was quite optimistically said. Now it reads: *"For EC-Earth, this adjustment corrects precipitation values to the direction of estimated ones, but it over-adjusts them for most of the simulated scenarios."*

We also added to the same paragraph the following text: *"It remains unclear why this temperature adjustment leads to an overestimation in the results for EC-Earth simulations. However, this could be related to the larger hydrological sensitivities for SAI compared to CO2 perturbations, as discussed in section 4.1.3. Although there are discrepancies between the actual simulated values and the estimated ones, this analysis shows that estimating the total precipitation change based…"*

**line 428: why is there the local radiative forcing peak at ≈50ºN and S? if I understand correctly, then the reference, Laasko 2020 sect. 3.1.2 explains nicely why the forcing effect is lower at the poles, but not why there is a local maximum between the subtropics and the poles.**

Now it reads: *"Furthermore, concerning stratospheric aerosols, the impact on radiative forcing is more pronounced at the Equator and latitudes around 50 degrees north and south where aerosol concentration is large due to the atmospheric circulation (Laakso et al. 2017). Thus radiative forcing is larger compared to the latitudes in between these regions"*

**Fig 7-10: how linear are the responses (within each model combination) at the local level? Is it possible, like you did on the global level, to understand the local response as approximately the sum of the slow response, fast GHG response and fast SAI response?**

In theory it is possible if column-integrated divergence of dry static energy is taken into account (see e.g. Zhang, S., Stier, P., and Watson-Parris, D.: On the contribution of fast and slow responses to precipitation changes caused by aerosol perturbations, Atmos. Chem. Phys., 21, 10179–10197, https://doi.org/10.5194/acp-21-10179-2021, 2021.). We conducted preliminary analyses on fast and slow responses at a regional level. However, as anticipated, the analysis encountered certain nonlinear complexities, making it less straightforward than a global-scale analysis. Consequently, local responses have been omitted from this study, but they may be explored in future research.

**≈ line 440, fig 8, CESM-SALSA, SRM20 and (in supplement S8) CESM-M7 SRM50: is there an AMOC response in the north Atlantic?**

This is now commented in the text: *"CESM simulations with larger CO2 concentration and large SAI injection rate (e.g SRM20-SALSA and SRM50-M7 (supplement Fig S8.)) are showing cooling in the North Atlantic which is associated to the weakening of the Atlantic Meridional Overturning Circulation seen also in simulations with global warming (Meehl et al., 2020; Fasullo and Richter, 2023)"*

**Regarding precip changes (fig 10): For impact modellers, maybe Precip-Evaporation would also be meaningful. Not sure if this is inside the scope of the paper. However, often SAI scenarios reduce not just precip but also evaporation, so that the overall effect on water availability is much less than precip changes suggest.**

We considered including the proposed figure in the supplement of the paper. However, as expected by the reviewer, it is a bit outside the scope of the paper and it is not easy to attach to the other analysis. Also, the study already has quite a lot of figures in the manuscript and it's supplement, so after consideration we decided to leave it out. However, it was a good suggestion for a possible future analysis and study.

**Last paragraph: Quantify "significant uncertainties". Is the inter-model discrepancy for the most relevant quantities (e.g., global precip change) of the order of 10% of the signal, or 50%, or is there even disagreement of the sign?**

Now it reads: *"The overall results of this study indicate that there are significant uncertainties regarding the estimated impacts of the possible deployment of SAI (e.g the coefficient of variation of the fast precipitation response was below injection rate 50 Tg(S)yr-1 were above 1.5)"*

**Minor Clarifications**

**fig. 4: Legend: the dash in "-SALSA" and "-M7" look like a minus, which is a little misleading. maybe write "for SALSA" or "(SALSA)" ? Also, please make more clear in the figure caption that in plots b-d, the symbols and the solid lines are independent, i.e., the solid line is the sum of the other lines (not: "total") whereas the symbols are the actual total (modelled) impacts. It becomes clear from the main text, but the figure itself is not as clear as it could be due to the shortness of the caption.**

Figure modified as suggested and in caption it now reads: *"The dashed line is precipitation change caused by SW absorption, the dash-dotted line is based on LW absorption and the solid line is the sum of these SW and LW components whereas markers are modelled fast precipitation responses from regression simulations."*

**Supplement figs S1, S2, S3, S5, maybe other equivalent ones: Please add unit to the y-axis label of plot a.**

Units added

**Fig. 6b: add "estimated" to the y-axis label (equiv to "modelled" in plot d). Clarify in caption that "hydrological sensitivity" (which is a quite general-sounding word with a much more specific meaning), refers to the effect of residual GMST change on precip.**

*"Predicted"* added to y-axis label and "estimated" changed to "predicted" in 6d. *"adjusted by hydrological sensitivity"* changed to **"***slow (temperature) response removed"*

**Section 5: you write a rather substantial summary of your findings. This could be further supported by adding references back to the corresponding sections and figures so that one can quickly (re)check the corresponding results in detail.**

References back to corresponding sections and figures are added.

Following typos and grammar are corrected as suggested:

**Typos, grammar  etc**

- **line 56: Changes … has -> have**
- **line 60: change translate -> translates**
- **line 254: sentence structure is a bit awkward**

Now it reads: *"The aforementioned observations emphasize that climate sensitivity is an idealized metric contingent on the timeframe considered"*

- **line 274-275: check sentence structure (missing "FOR larger injection…"?)**
- **line 291ff: Awkward sentence. Comma missing after "figure shows"?**
- **line 316: In case -> In this case?**
- **line 490: issensitive -> is sensitive**